# Kinetochore-independent chromosome segregation driven by lateral microtubule bundles

**Christina C Muscat[1†], Keila M Torre-Santiago[1†], Michael V Tran[1], James A Powers[2], Sarah M Wignall[1]\***

[1]Department of Molecular Biosciences, Northwestern University, Evanston, United States; [2]Light Microscopy Imaging Center, Indiana University, Bloomington, United States

**Abstract** During cell division, chromosomes attach to spindle microtubules at sites called kinetochores, and force generated at the kinetochore-microtubule interface is the main driver of chromosome movement. Surprisingly, kinetochores are not required for chromosome segregation on acentrosomal spindles in *Caenorhabditis elegans* oocytes, but the mechanism driving chromosomes apart in their absence is not understood. In this study, we show that lateral microtubule–chromosome associations established during prometaphase remain intact during anaphase to facilitate separation, defining a novel form of kinetochore-independent segregation. Chromosome dynamics during congression and segregation are controlled by opposing forces; plus-end directed forces are mediated by a protein complex that forms a ring around the chromosome center and dynein on chromosome arms provides a minus-end force. At anaphase onset, ring removal shifts the balance between these forces, triggering poleward movement along lateral microtubule bundles. This represents an elegant strategy for controlling chromosomal movements during cell division distinct from the canonical kinetochore-driven mechanism.

**\*For correspondence:** s-wignall@northwestern.edu

†These authors contributed equally to this work

**Competing interests:** The authors declare that no competing interests exist.

## Introduction

To facilitate partitioning of the genome during cell division, dynamic spindle microtubules capture chromosomes at sites called kinetochores, forming end-on attachments. These attachments are extremely important, as they allow chromosomes to harness the forces generated by microtubule dynamics to drive movements crucial for proper segregation (*Rago and Cheeseman, 2013*; *Cheerambathur and Desai, 2014*; *Cheeseman, 2014*). During prometaphase, chromosomes use these forces to align at the center of the spindle in a process called congression and subsequently in anaphase, kinetochores couple separating chromosomes to depolymerizing microtubules, powering segregation (*Walczak and Heald, 2008*; *Kops et al., 2010*). Despite this fundamental role for the kinetochore in modulating chromosome dynamics, a recent study reported the surprising discovery that kinetochores are not required for chromosome segregation in *Caenorhabditis elegans* female reproductive cells (oocytes); the authors found that kinetochore components are removed from chromosomes during anaphase, and when they perturbed kinetochore assembly, chromosomes still segregated relatively normally (*Dumont et al., 2010*). The mechanisms driving chromosomes apart in *C. elegans* oocytes are not understood but are important to uncover, as they are a departure from the canonical kinetochore-driven mode of segregation and therefore represent a new strategy for controlling chromosome dynamics during cell division.

The divisions of oocytes differ from other cell types in two major respects. First, reproductive cells undergo a single round of DNA replication followed by two meiotic divisions where homologous

**eLife digest** An animal's genetic material is packaged into structures called chromosomes. Most animals have two sets of chromosomes: one from each parent. Sperm and egg cells must contain half the number of chromosomes compared to other cells in the body, so that when they fuse, the resulting embryo receives a full complement of chromosomes.

Egg and sperm cells are made via a type of cell division called meiosis. In meiosis, the genetic material of a cell is copied once but then the cell divides twice. Therefore, at the end of the two divisions, the resulting sperm or egg cells contain half the number of chromosomes as the original cell. During cell division, the genetic material is separated by a structure called the spindle apparatus.

The spindle is made of protein filaments called microtubules. At each end of the spindle, there is a cluster of microtubule ends, known as a 'pole'. The other ends of the microtubules extend out towards the center of the spindle, where they overlap with the microtubules from the opposite pole. The chromosomes line up in the center of the spindle and then the chromosomes are separated, with half moving to one spindle pole, and half to the other. In most forms of cell division, the microtubules attach to the chromosomes via sites called kinetochores. However, it was recently discovered that kinetochores are not required to separate chromosomes to make egg cells in the worm *C. elegans*, suggesting that these chromosomes associate with the spindle in a different way.

Muscat, Torre-Santiago et al. have now used high-resolution imaging to look at this chromosome separation process in more detail and to figure out how the chromosomes separate when *C. elegans* forms egg cells. The experiments revealed that the chromosomes move within the spindle along parallel microtubule bundles, much like trains moving along a track. The chromosomes are moved into position at the center of the spindle by a ring-shaped group (or 'complex') of proteins that forms around the center of each chromosome. The protein complex comes off the chromosomes as they separate, and a motor protein called dynein walks along the microtubules to pull the separated chromosomes to the poles.

Muscat, Torre-Santiago et al.'s findings thus show that meiosis in *C. elegans* during the production of egg cells works in a very different way to other types of cell division. In the future, it will be important to understand how dynein and the ring-shaped complex are regulated, as this may shed light on what causes mistakes in the separation of genetic material during meiosis, which can lead to infertility, miscarriages, and birth defects in humans and other animals.

chromosomes (in Meiosis I) and then sister chromatids (in Meiosis II) segregate away from one another, generating haploid gametes. Second, oocytes of most species lack centrosomes, which in other cells duplicate and then nucleate and organize microtubules, forming the two spindle poles (*Manandhar et al., 2005*). Therefore, microtubules are not nucleated from pre-defined poles in oocytes and instead are sorted and organized into a bipolar structure around the chromosomes (*Dumont and Desai, 2012*). This difference influences the way chromosomes initially associate with spindle microtubules and the mechanism by which they achieve metaphase alignment. Work in mouse oocytes has demonstrated that homologous chromosome pairs (bivalents) lack end-on kinetochore-microtubule interactions during prometaphase (*Brunet et al., 1999*), and stable attachments are not achieved until they have already congressed to the metaphase plate (*Kitajima et al., 2011*). Similarly, we previously found that *C. elegans* oocytes utilize a unique congression mechanism. *C. elegans* chromosomes are holocentric, so kinetochore proteins form cup-like structures around the ends of bivalents (in Meiosis I) and sister chromatid pairs (in Meiosis II) (*Monen et al., 2005*) (*Figure 1A*, orange). However, we found that microtubule density is low at chromosome ends in prometaphase and instead lateral microtubule bundles run along the sides of chromosomes (*Figure 1A*, green). We also found that the microtubule motor KLP-19 forms a ring around the center of each bivalent (in Meiosis I) and at the sister chromatid interface (in Meiosis II) (*Figure 1A*, red) and provides a plus-end directed force on the chromosomes. Therefore, chromosome movement on the *C. elegans* acentrosomal spindle during congression is mediated by movement of chromosomes along laterally associated microtubule bundles, facilitated by plus-end directed forces operating on the chromosomes (*Wignall and Villeneuve, 2009*). The discovery that chromosome segregation in these cells does not require kinetochores (*Dumont et al., 2010*) raised the possibility that lateral

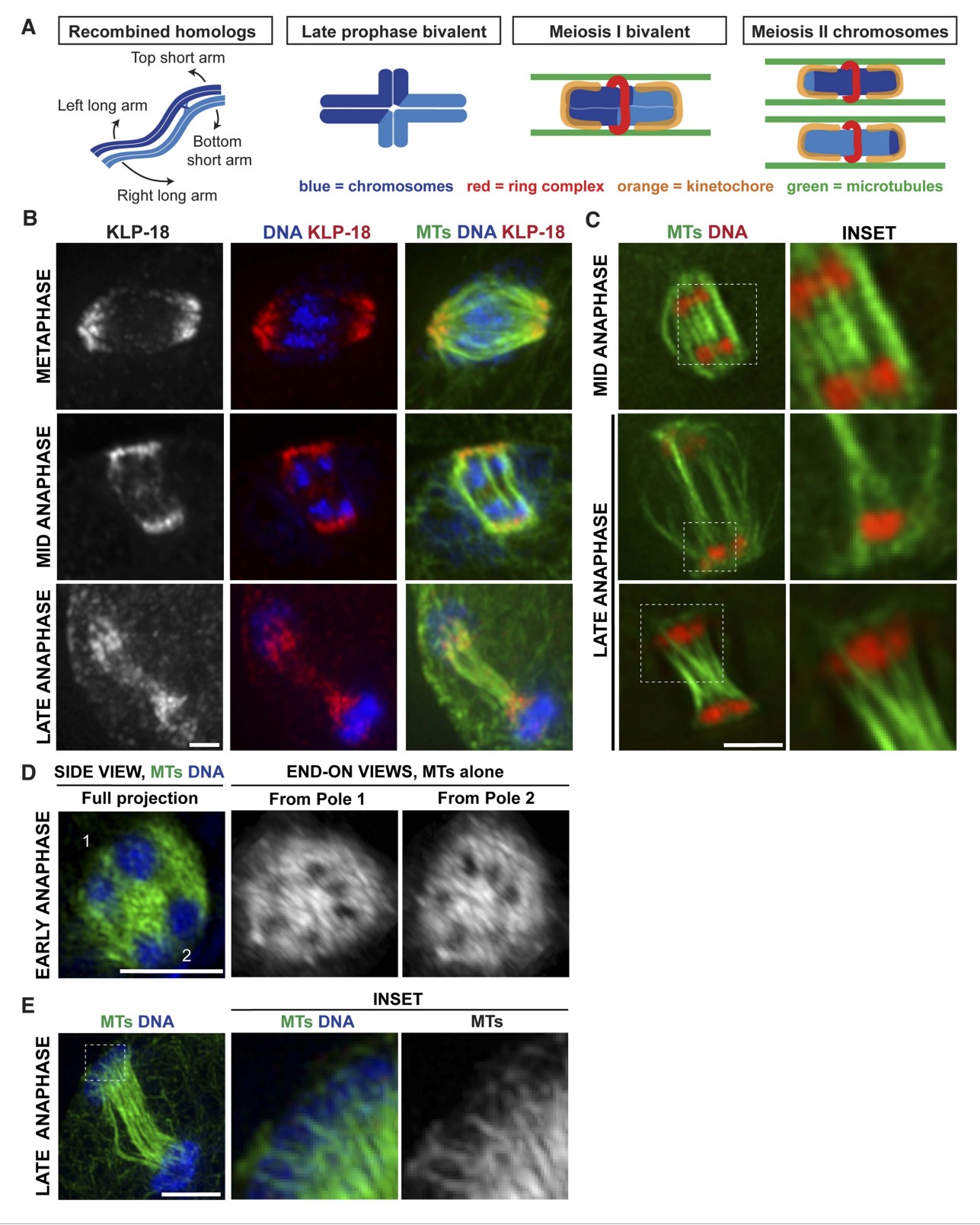

**Figure 1.** Poles broaden but remain intact during anaphase, creating open channels. (**A**) Chromosome organization during *C. elegans* meiosis. Homologous chromosomes (blue) undergo recombination to link them together and then reorganize to form a cruciform bivalent with long and short arms. This structure condenses further and a holocentric cup-like kinetochore (orange) forms around the bivalent ends and a complex of proteins (red)

*Figure 1. continued on next page*

*Figure 1. Continued*

forms a ring around the center. Following homolog separation, the two resulting chromosomes each form a similar structure, with kinetochores cupping the chromosome ends and a ring around the sister chromatid interface. Bundles of microtubules (green) associate laterally with chromosomes during both meiotic divisions. (**B**) Oocyte spindles stained for tubulin (green), DNA (blue), and spindle pole marker KLP-18 (red). KLP-18 is at focused poles in metaphase (top) and remains as poles broaden in mid-anaphase (middle). Chromosomes move past the poles in late anaphase (bottom). (**C**) Live imaging of worms expressing mCherry-histone (red) and GFP-tubulin (green). Full projections of entire spindles are shown on the left and partial projections of insets chosen to highlight particular features of spindle organization are on the right. Lateral bundles associate with and extend beyond segregating chromosomes in mid-anaphase (top row) and maintain association with chromosomes as they move past the poles (bottom). (**D**) Super-resolution image of an early anaphase oocyte spindle stained for DNA (blue) and tubulin (green). End-on views show a rotated 3D rendering and highlight the six open channels between segregating chromosomes. (**E**) Super-resolution image of a late anaphase oocyte spindle stained for DNA (blue) and tubulin (green) showing microtubule bundles splitting around the chromosomes as they reach microtubule ends. Bars = 2.5 μm.

microtubule associations might also drive chromosome separation, which would represent a new strategy for segregation.

Here, we use super-resolution microscopy to reveal that chromosomes remain associated with lateral microtubule bundles during anaphase on *C. elegans* acentrosomal spindles, and we define a mechanism that facilitates segregation in the context of this organization. We find that a balance of forces mediates chromosome dynamics during congression and segregation, where plus-end directed forces originating in the rings are countered by progressive accumulation of minus-end directed forces on the chromosomes. Removal of the ring at the metaphase to anaphase transition then shifts this balance, triggering poleward movement. Therefore, we have defined a novel mode of chromosome segregation in oocyte meiosis, facilitated by opposing chromosomal forces driving movement along lateral microtubule bundles.

## Results

### Microtubules in the oocyte spindle reorganize in anaphase to create channels that are open from pole to pole

When chromosome separation in *C. elegans* oocytes was demonstrated to be kinetochore-independent, a model was proposed that spindle poles disassemble at anaphase onset and a new microtubule array polymerizes between segregating chromosomes to push them apart; an outward-pushing force generated by these microtubules on the inside surfaces of chromosomes was suggested to be the primary candidate for driving separation (*Dumont et al., 2010*). To test this model, we used high-resolution imaging to visualize a component of acentrosomal spindle poles as the spindle reorganized during anaphase (*Figure 1B*). Consistent with previous studies, we found that a number of events occur concomitantly at the metaphase to anaphase transition: (1) the spindle rotates until it is perpendicular to the cortex, (2) the spindle shrinks dramatically, and (3) the poles broaden. Then, as anaphase progresses, the spindle elongates again (*Albertson and Thomson, 1993*; *Yang et al., 2003*; *McNally and McNally, 2005*; *McNally et al., 2006*). However, the spindle pole marker KLP-18 remained associated with microtubules throughout anaphase progression despite these morphological changes; the chromosomes moved towards the KLP-18-marked poles and then moved past them in late anaphase (*Figure 1B*). This finding is consistent with earlier work showing that another pole marker, ASPM-1, also remains at poles during anaphase (*van der Voet et al., 2009*; *McNally and McNally, 2011*) and with previous lower resolution imaging of KLP-18 localization (*Segbert et al., 2003*). Therefore, spindle poles broaden but remain intact during anaphase, suggesting that the metaphase microtubule array is reorganized but does not disassemble.

Since we previously showed that lateral microtubule bundles run alongside chromosomes during congression (*Wignall and Villeneuve, 2009*) (*Figure 1A*), we wanted to determine whether these lateral contacts were present during anaphase, potentially to facilitate segregation. High-resolution live imaging of oocytes expressing GFP-tubulin and mCherry-histone (to mark microtubules and chromosomes, respectively) revealed parallel bundles of microtubules running alongside and extending beyond pairs of separating chromosomes during early- and mid-anaphase (*Figure 1C*). In partial projections (focused on particular sets of separating chromosomes), bundles of microtubules could be observed on both sides of the chromosomes, with a lower density of microtubules in

between the separating pair (*Figure 1C*, top inset), suggesting that chromosomes may move through 'channels' as they segregate. To better visualize this organization, we used super-resolution imaging to obtain a stack of images through the entire spindle, then made a 3D rendering and rotated the image to achieve end-on views from each pole. This imaging revealed six regions of low-microtubule density in the anaphase spindle, representing the six sets of separating homologous chromosomes in *C. elegans*; this organization was clear in both early- (*Figure 1D*, *Video 1*) and mid-anaphase (*Figure 2A*, *Videos 2–5*). Therefore, each separating chromosome pair sits within its own channel in the microtubule array, and each of these channels is open from pole to pole. In late anaphase, chromosomes are found at the extreme ends of the microtubules at the spindle poles and the microtubule bundles in the center of the spindle converge, closing the channels (*Figure 1C,E*). However, even at this stage, lateral microtubule bundles maintain contact with the chromosomes, splitting around them at the poles (*Figure 1C,E*, insets). Taken together, these results reveal that chromosomes associate with lateral bundles of microtubules during anaphase and are inconsistent with a model in which a new array of microtubules polymerized between separating chromosomes provides a pushing force on their inside surfaces to drive separation.

## Rings dissociate from chromosomes in anaphase and are found in the microtubule channels

We previously found that a set of proteins forms a ring around the mid-region of each chromosome pair during congression (*Figure 1A*); this ring includes the kinesin motor KLP-19 and the chromosome passenger complex (CPC), which contains the kinase AIR-2/Aurora B (*Wignall and Villeneuve, 2009*). While AIR-2/Aurora B colocalizes with microtubules in anaphase (*Schumacher et al., 1998*; *Speliotes et al., 2000*; *Romano et al., 2003*), KLP-19 and a few other ring components stay as rings and then elongate into bar-shaped structures between separating chromosomes (*Dumont et al., 2010*). We used super-resolution imaging to further assess the morphology of these structures and to examine their organization in relation to the microtubule channels. First, we found that the MPM-2 antibody (raised against mitotic phospho-proteins [*Davis et al., 1983*]), which had been previously shown to stain the mid-region of meiotic bivalents (*Kitagawa and Rose, 1999*), marked the ring structures in both metaphase and anaphase (*Figure 2A–C*). During anaphase, the MPM-2-marked rings completely dissociated from chromosomes and remained in the center of the spindle (*Figure 2B*). In mid-anaphase, these structures appeared elongated but retained a ring-like appearance (*Figure 2B*, asterisk), suggesting that the rings flatten as anaphase progresses, forming the bar-shaped structures previously described (*Dumont et al., 2010*). Importantly, these structures were located within the six microtubule channels; each open channel contained a pair of separating chromosomes with a disassembling ring between them (*Figure 2A*, *Videos 2–5*). In late anaphase, at the stage where microtubule bundles in the central spindle converge (*Figure 1C*), MPM-2 staining was diffuse and other ring components were also not discernable as distinct structures, suggesting that the rings had disassembled (*Figure 2—figure supplement 1*).

To determine the timing of ring release from chromosomes, we next co-stained with MPM-2 and an antibody against separase (SEP-1), a protease that cleaves a subunit of the cohesin complex to release sister chromatid cohesion and allow homologous chromosome separation (*Siomos et al., 2001*; *Kudo et al., 2006*). We found that when the rings were associated with chromosomes in metaphase, SEP-1 localized to

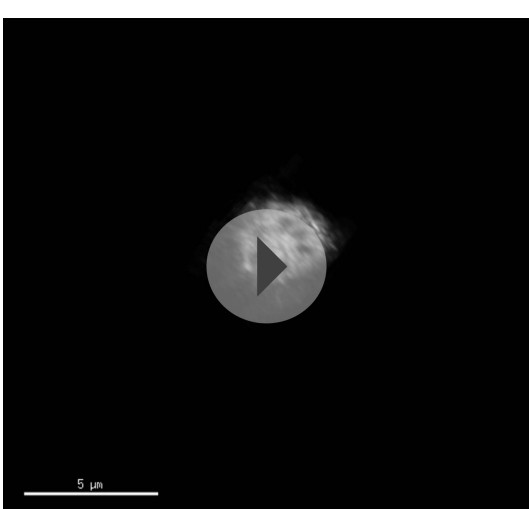

5 µm

**Video 1.** Super-resolution imaging revealing six open channels in the early anaphase spindle. Super-resolution image of a fixed oocyte spindle stained for tubulin. Video is a rotation of a 3D volume rendering generated using Softworx software (Applied Precision).

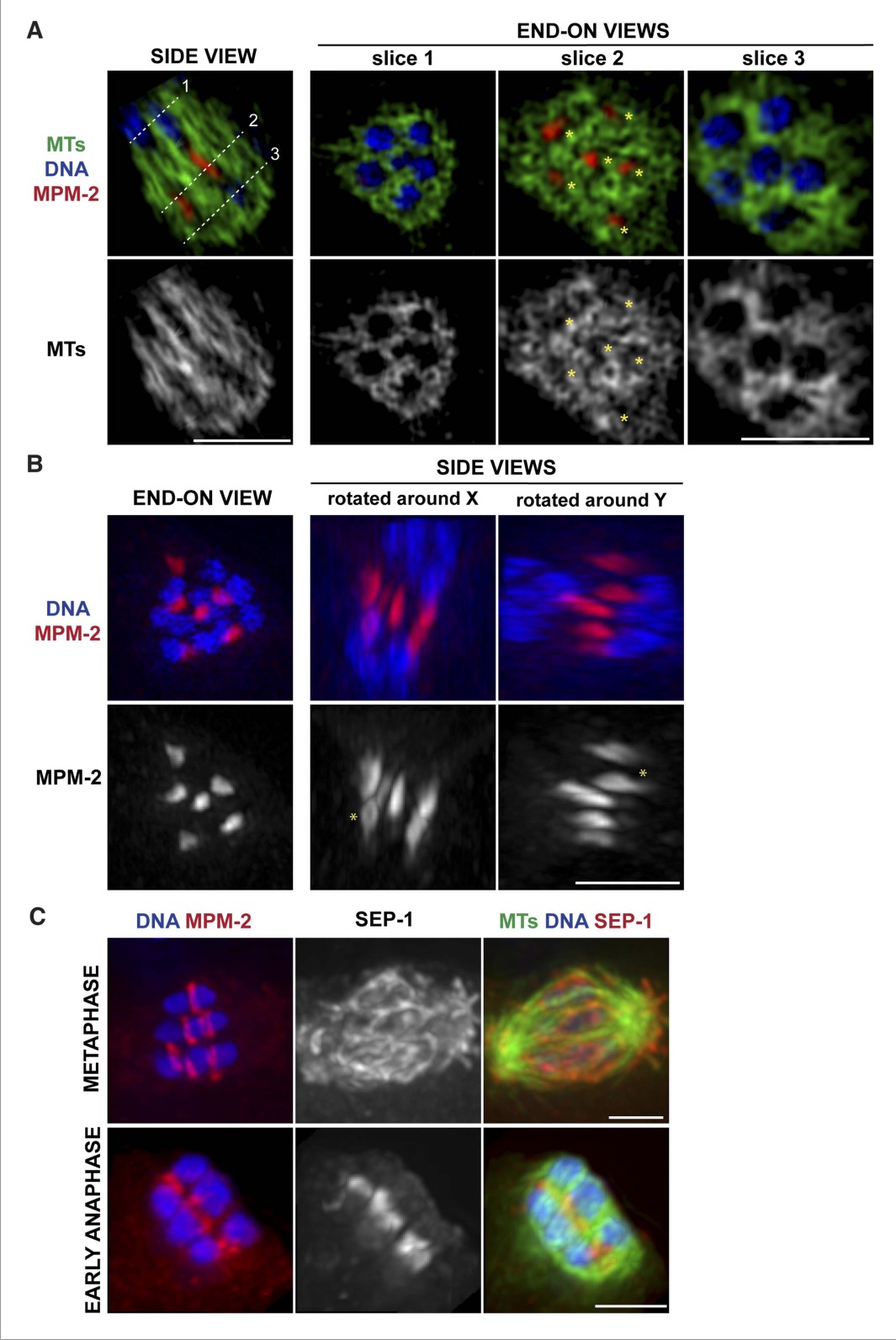

**Figure 2**. Ring structures are located within the microtubule channels during anaphase. (**A**, **B**) Super-resolution imaging of an anaphase oocyte spindle stained for DNA (blue), tubulin (green), and ring marker MPM-2 (red). (**A**) The first column shows a single z-slice highlighting an open channel containing a separating chromosome pair with a flattened ring in between. Columns 2, 3, and 4 are end-on views (each a single z-slice, depicting the region

*Figure 2. continued on next page*

*Figure 2. Continued*

denoted by the corresponding dotted line on the first image), showing the chromosomes and disassembling rings within microtubule channels. (**B**) Rings dissociate from the chromosomes; although most of the structures appear flattened, partial rings can still be observed ('asterisks'). (**C**) Oocyte spindles stained for DNA (blue), ring marker MPM-2 (red in left column), SEP-1/separase (middle column, red in right column), and tubulin (green). SEP-1 localizes to cup-like structures and filaments within the spindle during metaphase, consistent with kinetochore localization (top row). In anaphase (bottom row), SEP-1 localizes to the rings. Bars = 2.5 μm.

The following figure supplement is available for figure 2:

**Figure supplement 1**. Ring structures are gone in late anaphase.

cup-like structures surrounding bivalents and to filamentous structures found within the spindle (*Figure 2C*), a pattern also exhibited by meiotic kinetochore components (*Monen et al., 2005*), consistent with an earlier study (*Bembenek et al., 2007*). However, during anaphase, SEP-1 localized to the ring structures; this dramatic relocalization correlated with anaphase onset, as SEP-1 was found on the rings in early anaphase, when the spindle had rotated but chromosomes had barely separated (*Figure 2C*). Therefore, separase likely redistributes to the mid-region of meiotic chromosomes at anaphase onset to cleave cohesin and then remains associated with the ring structures during anaphase, suggesting that ring removal from chromosomes is coordinated with cohesion release.

## Chromosomes move towards microtubule minus ends during monopolar anaphase

Given the organization of the anaphase spindle and the finding that kinetochores are not required for chromosome segregation (*Dumont et al., 2010*), we hypothesized that segregation could be mediated by minus-end directed movement of chromosomes along lateral microtubule bundles. To examine this idea, we first assessed chromosome behavior during anaphase after inducing monopolar spindle formation; we previously showed that microtubule bundles make lateral contacts with chromosomes in both bipolar and

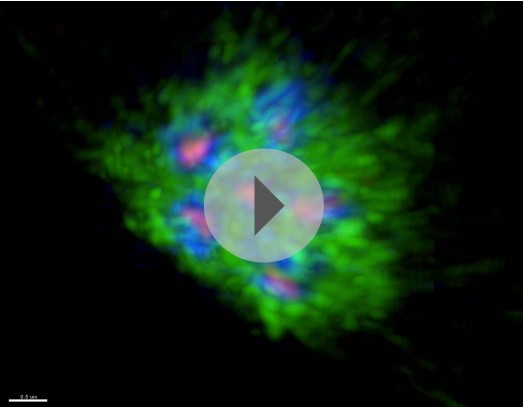

**Video 2.** Super-resolution view of spindle organization in mid-anaphase. Super-resolution image of a fixed oocyte spindle stained for DNA (blue), tubulin (green), and ring marker MPM-2 (red). Video shows a 3D volume rendering, generated using Imaris software (Bitplane) and the Orthogonal Slice View function is used to show individual slice views within the spindle. Individual components blink in and out to highlight particular features of spindle organization; specifically that each pair of separating chromosomes is located within an open microtubule channel, with a disassembling ring structure in between.

**Video 3.** Super-resolution view of spindle organization in mid-anaphase. Super-resolution image of a fixed oocyte spindle stained for DNA (blue), tubulin (green), and ring marker MPM-2 (red). This is the same video as *Video 2*, but without individual components blinking in and out.

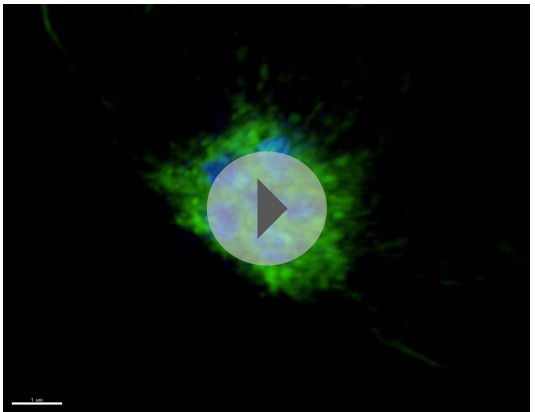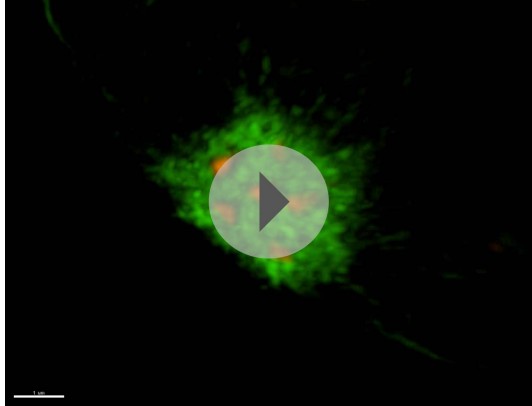

**Video 4.** Super-resolution view of spindle organization in mid-anaphase. Super-resolution image of a fixed oocyte spindle stained for DNA (blue) and tubulin (green). This is the same sequence as *Video 3*, but with only DNA and tubulin displayed.

**Video 5.** Super-resolution view of spindle organization in mid-anaphase. Super-resolution image of a fixed oocyte spindle stained for tubulin (green) and ring marker MPM-2 (red). This is the same sequence as *Video 3*, but with only tubulin and the rings displayed.

monopolar spindles (*Wignall and Villeneuve, 2009*), enabling us to compare chromosome behavior in these two contexts. Assessing monopolar anaphase is particularly interesting because the organization of microtubules in these spindles (with minus ends in the center and plus ends out) makes them ideal for dissecting the contributions of plus- and minus-end-directed forces. In this case, if minus-end-directed movement drives segregation we would expect to observe movement of separated chromosomes towards the central pole. Live imaging of monopolar spindle dynamics demonstrated that in Meiosis I the six bivalents move away from the central pole, oscillate, and then coordinately move inwards to the pole (*Figure 3A*, *Video 6*); in most cases, polar body extrusion failed so all chromosomes were retained in the cell. This behavior was then repeated in Meiosis II, with the 12 separated chromosomes moving out as the monopolar spindle forms, oscillating, and then moving back in (*Figure 3B*, *Video 7*). Poleward movement of chromosomes in monopolar spindles has been previously described and was interpreted to represent capture of chromosomes by microtubules followed by a congression-like process that drew them to the center (*Connolly et al., 2014*). However, we found that separase colocalizes with ring components when the chromosomes are located near the central pole (*Figure 3C*), demonstrating that this configuration represents anaphase and not congression.

Though our results are consistent with the idea that separated chromosomes display minus-end directed movement on monopolar spindles, because the anaphase spindle shrinks (*Albertson and Thomson, 1993*; *Yang et al., 2003*; *McNally and McNally, 2005*; *McNally et al., 2006*) another possibility is that chromosomes move towards the central pole as a result of the microtubules shortening and the chromosomes moving inwards passively. To investigate this possibility, we induced monopolar spindle formation in a strain that had also been depleted of the microtubule depolymerizing kinesin MCAK/KLP-7, as we found that MCAK depletion increased spindle size in both metaphase (*Figure 3—figure supplement 1*) and anaphase (*Figure 3D*). Following MCAK depletion, monopolar spindles remained large in anaphase, but chromosomes still moved in (*Figure 3D*, top row), indicating that spindle shrinkage and microtubule depolymerization is not responsible for the inward movement. Importantly, we found that while chromosomes move inwards in monopolar anaphase, the ring structures could be observed out towards the periphery of the monopolar spindle, where the plus ends are located. We obtained the same result without MCAK depletion; although these monopolar spindles were smaller due to spindle shortening, the ring structures could still be found near the microtubule plus ends when the chromosomes were found at the central pole (*Figure 3C,D*). Therefore, similar to bipolar anaphase, where the rings are left behind in the center of the spindle, in monopolar anaphase they are also removed from chromosomes and remain near the plus ends, presumably where the chromosomes were located when anaphase was induced.

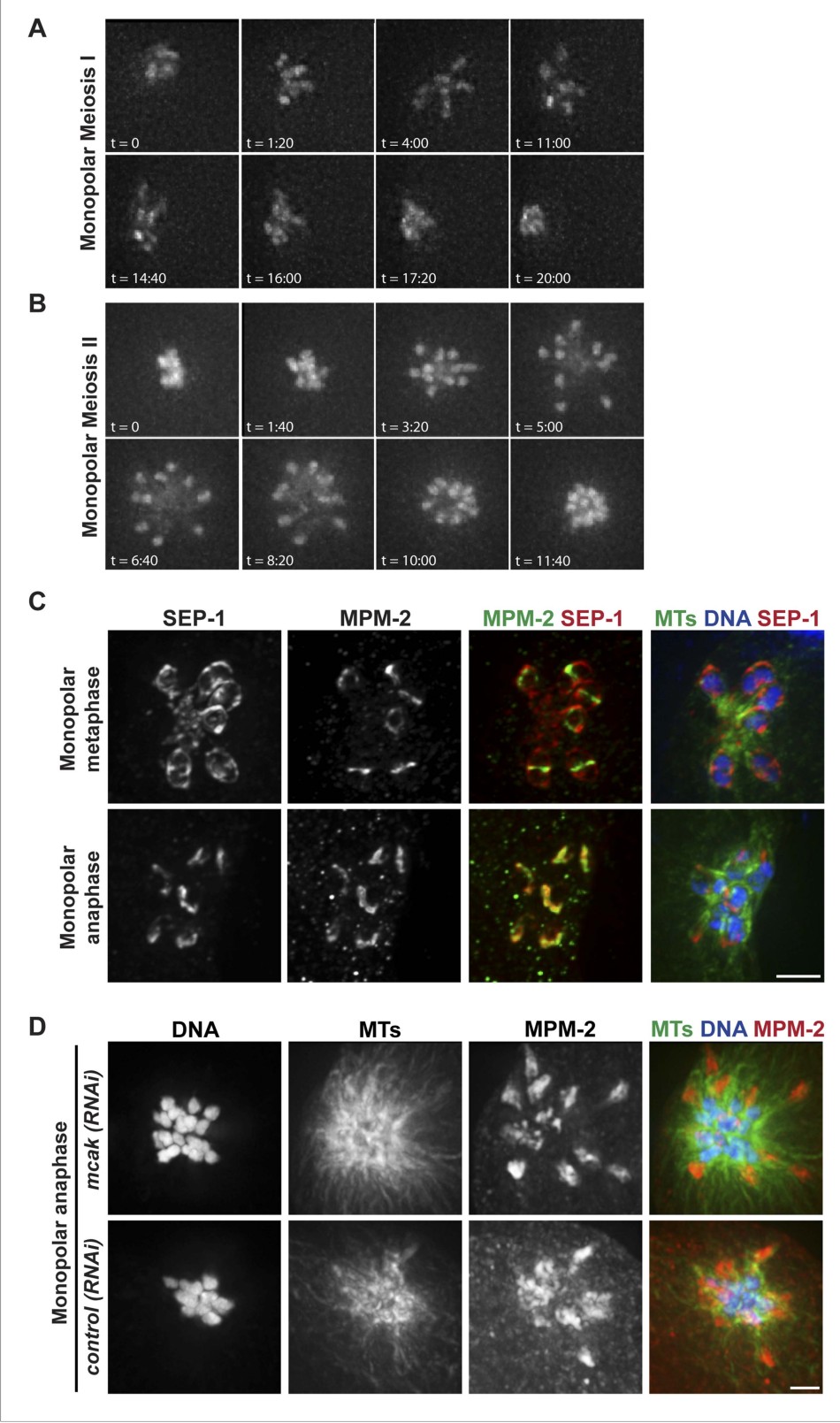

**Figure 3**. Chromosomes move towards microtubule minus ends during monopolar anaphase. (**A**, **B**) Montages of time-lapse imaging of a strain expressing GFP-histone; GFP-tubulin following *klp-18 (RNAi)* to generate monopolar spindles during Meiosis I (**A**) and II (**B**). Chromosomes move away from the pole during monopolar spindle formation
*Figure 3. continued on next page*

*Figure 3. Continued*

and then coordinately move polewards during anaphase. (**C**) *klp-18 (RNAi)* monopolar oocyte spindles stained for DNA (blue), SEP-1 (red), ring marker MPM-2 (green in third column), and tubulin (green in fourth column). When chromosomes are away from the pole in metaphase, SEP-1 localizes to kinetochores and then moves to the ring structures in anaphase. (**D**) *klp-18 (RNAi)* monopolar oocyte spindles stained for tubulin (green), DNA (blue), and ring marker MPM-2 (red). In the top row, MCAK/KLP-7 was also depleted, creating a larger spindle. When separated chromosomes move to the pole in monopolar anaphase, the rings stay out. Bars = 2.5 µm.

The following figure supplement is available for figure 3:

**Figure supplement 1**. MCAK inhibition causes larger monopolar spindles.

## Minus-end directed forces act on oocyte chromosomes prior to anaphase

Our data so far are consistent with the idea that during anaphase in oocyte spindles, minus-end directed forces drive chromosome movement through open microtubule channels to spindle poles. This would represent a switch in the dominant force operating on chromosomes since during congression, plus-end directed forces (including KLP-19 in the ring [*Wignall and Villeneuve, 2009*]) promote movement to the metaphase plate. To better understand how this switch occurs, we next wanted to determine whether minus-end forces only act on chromosomes in anaphase, or whether they also act on chromosomes during congression, counterbalancing the plus-end forces. Therefore, we imaged monopolar spindles that had been arrested in metaphase using a temperature-sensitive mutant in the anaphase promoting complex (APC) (*emb-27ts*); following shift to the restrictive temperature, oocytes continue to be ovulated, but then arrest at metaphase I (*Golden et al., 2000*). Therefore, fixed imaging yields a mixture of oocytes that have been arrested for varying amounts of time. Intriguingly, we observed multiple phenotypes in our experiments, potentially reflecting differences in the length of APC arrest. First, a significant fraction of structures (48/119) looked

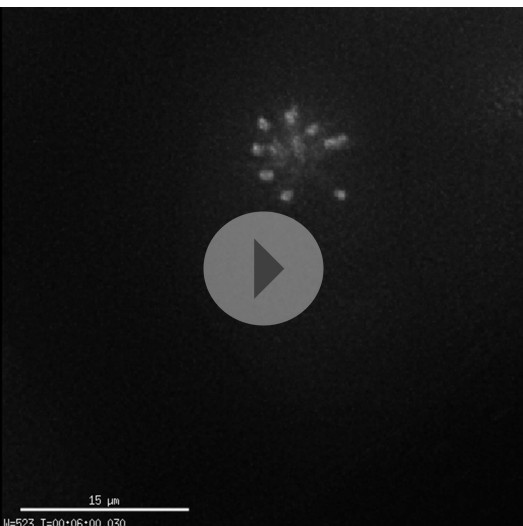

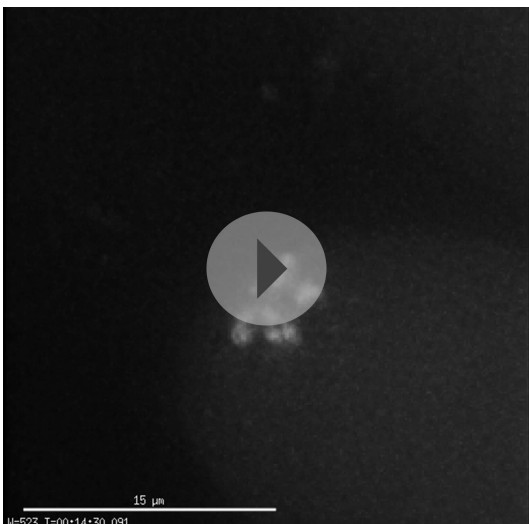

**Video 6.** Dynamics of monopolar Meiosis I. Time-lapse imaging of a strain expressing GFP-histone; GFP-tubulin following *klp-18 (RNAi)* to generate monopolar spindles. In monopolar Meiosis I, the 6 bivalents move away from the central pole, oscillate, and then coordinately move inwards to the pole. Frames were acquired every 10 s and displayed at 5 frames per second.

**Video 7.** Dynamics of monopolar Meiosis II. Time-lapse imaging of a strain expressing GFP-histone; GFP-tubulin following *klp-18 (RNAi)* to generate monopolar spindles. In monopolar Meiosis II, the 12 separated chromosomes move away from the central pole, oscillate, and then coordinately move inwards to the pole. Frame acquisition same as *Video 6*.

indistinguishable from unarrested prometaphase/metaphase monopolar spindles, with bivalents located away from the central pole and a ring around the center of each (*Figure 4A*, top). In these monopolar spindles, chromosomes tended to be oriented such that the long axis of the bivalent was parallel to the direction of the microtubule bundles, as they are found in unarrested monopolar spindles (*Wignall and Villeneuve, 2009*). However, we also observed many monopolar spindles (42/119) where the rings appeared dramatically stretched in the direction of the microtubule plus ends (*Figure 4A*). Some rings appeared to be stretching off the bivalents on both sides (arrows), while others were predominantly stretched off one side (arrowheads); in the latter case, the bivalents were sometimes turned 90° from their usual orientation. These observations are consistent with the interpretation that under metaphase arrest, a plus-end directed force acts on the rings, causing them to stretch, and a counterbalancing minus-end force is exerted on the chromosome, resisting the stretching force; when the ring predominantly stretches off of one side of the bivalent, this tug-of-war can cause the bivalents to rotate. As further evidence that minus-end-directed forces act on chromosomes under metaphase arrest, we also observed structures that had chromosomes at the center of the microtubule aster similar to normal monopolar anaphase (29/119); in most of these cases, the rings had come off the chromosomes and were located near the microtubule plus ends, though in a minority of cases the rings were dramatically stretched but still on the chromosomes. Inward chromosome movement was not the result of premature release from the metaphase arrest, as the kinetochore component KNL-3 still localized around the bivalents in these structures (*Figure 4B*) (despite the fact that kinetochore components are normally removed in anaphase [*Monen et al., 2005*]), and two-cup like kinetochores could be observed on each bivalent, confirming that the homologous chromosomes had not separated and the cells had not initiated anaphase. Moreover, the rings often appeared broken (forming fragments of varying sizes, *Figure 4A*, asterisks), suggesting that they had been aberrantly removed from bivalents without the release of cohesion. Taken together, these results indicate that minus end forces act on chromosomes prior to anaphase induction. The fact that we observe a mixed population of structures suggests that these forces accumulate under metaphase arrest, causing the rings to stretch and then ultimately be removed from the chromosomes as these increasing forces drive chromosomal movement to minus ends.

## Dynein contributes to chromosome segregation on oocyte spindles

Next, we wanted to identify factors that contribute to the minus-end force operating on chromosomes. As dynein is an important minus-end-directed motor that plays diverse roles during mitosis and meiosis (*Raaijmakers and Medema, 2014*), we set out to determine whether it could also help drive chromosome movement on acentrosomal spindles. Previously, dynein has been shown to localize diffusely to the oocyte spindle during prometaphase and metaphase and then further concentrate at poles as the spindle rotates in preparation for anaphase (*Ellefson and McNally, 2009*; *van der Voet et al., 2009*). We confirmed this localization pattern, but we also detected a population of dynein that cupped meiotic chromosomes. This chromosomal pool of dynein was difficult to distinguish on prometaphase and metaphase bipolar spindles, potentially due to the fact that dynein localized all over the spindle as well, which could obscure a faint chromosomal pool. However, chromosomal localization was more clearly visible on monopolar spindles. Since chromosomes are spread out in these structures, the spindle pool of dynein was more separated from the chromosomal pool, allowing visualization of dynein cupping the bivalent ends (*Figure 5A*). On these chromosomes, dynein was often seen asymmetrically localized, with stronger staining on the end of the bivalent pointing outwards towards the microtubule plus ends (asterisks), raising the possibility that dynein may preferentially load from this direction. However in many cases, dynein could be observed on both faces of the chromosomes (arrowheads), suggesting either that dynein loading is not always asymmetric, or that asymmetrically loaded dynein can ultimately spread to the entire chromosomal surface. In anaphase, we observed dynein at the poleward surfaces of separating chromosomes, again cupping the chromosomal surface (*Figure 5B*), but undetectable on the inside surfaces (in the region where chromosomes would have been previously associated before release of cohesion). These findings are in line with previous work that reported that dynein moves from spindle poles towards the outside surfaces of chromosomes in anaphase (*Dumont et al., 2010*). Therefore, dynein localizes to meiotic chromosomes both prior to and during anaphase, placing it in a location where it could provide a minus-end directed force during congression and segregation.

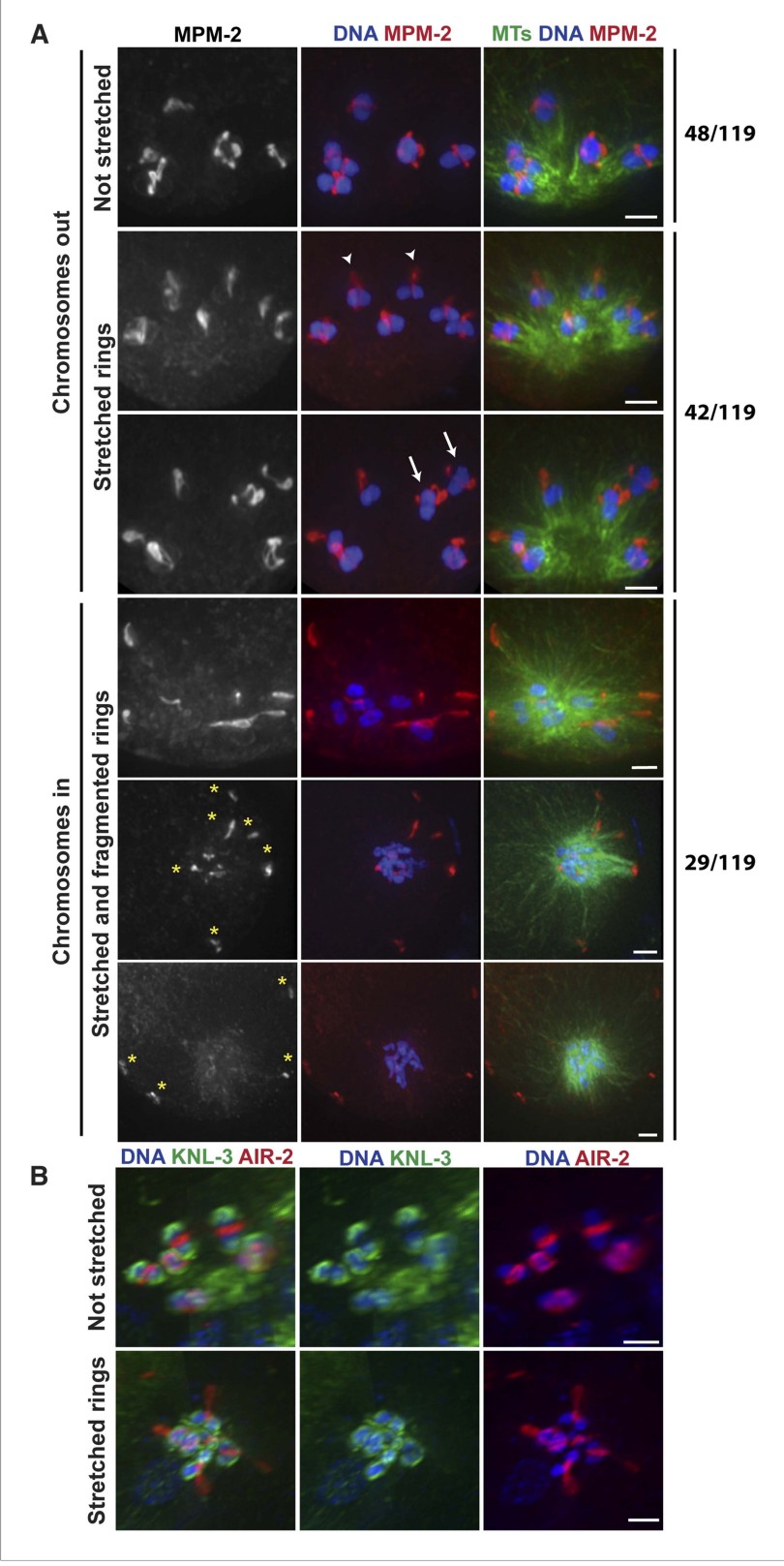

**Figure 4**. Chromosomes are subjected to minus-end directed forces prior to anaphase. (**A**) Monopolar oocyte spindles arrested at Metaphase I stained for DNA (blue), tubulin (green), and ring marker MPM-2 (red). Top row shows intact rings. Rows 2 and 3 show examples of rings stretching, but still near chromosomes ('arrowheads' show

*Figure 4. continued on next page*

*Figure 4. Continued*

rings predominantly coming off one side of the bivalent, 'arrows' show rings off both sides). Bottom three rows show examples of chromosomes at the pole and ring fragments ('asterisks') at microtubule plus ends or off the spindle. (**B**) Monopolar oocyte spindles arrested at Metaphase I stained for DNA (blue), kinetochore protein KNL-3 (green), and ring component AIR-2 (red). The kinetochore remains intact in spindles with stretched rings, indicating that the cells have not initiated anaphase. Bars = 2 μm.

To test whether dynein functionally contributes to chromosome movements, we used multiple inhibition strategies and then assessed the effects on chromosome positioning and segregation. These experiments were complicated by the fact that full dynein inhibition (either by strong *dhc-1* RNAi or high concentrations of the dynein inhibitor ciliobrevin [*Firestone et al., 2012*]) resulted in spindle defects (*Figure 5—figure supplement 1*). Therefore, we used partial or conditional dynein inhibition, first finding conditions where spindle morphology looked normal and then assessing chromosome behavior on those spindles.

To determine if dynein provides a minus-end directed force on chromosomes, we first assessed whether partial dynein depletion altered chromosome position on monopolar spindles. During normal monopolar spindle formation, chromosomes move out away from the pole and oscillate (*Figure 3A,B*, *Videos 6*, *7*); in fixed images, we found that 34% of chromosomes were found near the plus ends of microtubules, reflecting this dynamic positioning (*Figure 5C*, gray bars). However, significantly more chromosomes (60%) were found in this location following dynein depletion (*Figure 5C*, black bars), suggesting that dynein generates a minus-end directed force on chromosomes that affects their positioning. Consistent with this finding, we also observed lagging chromosomes on anaphase monopolar spindles following partial dynein inhibition. While chromosomes usually move coordinately to the pole in monopolar anaphase (*Figure 3A,B*), a significant fraction of spindles had some chromosomes towards microtubule plus ends upon treatment with low concentrations of ciliobrevin (*Figure 5D*; 15/21 monopolar anaphases), suggesting that dynein provides a minus-end directed force during anaphase as well as during congression.

Next, we assessed the effects of dynein inhibition on chromosome segregation on bipolar spindles. To reduce the possibility that lagging chromosomes were the result of altered spindle morphology, we took advantage of a temperature-sensitive mutant (*dhc-1(ct76)*). A previous study reported that dynein is inactivated within a few minutes in this mutant and used short temperature shifts show that dynein plays a role in mitotic chromosome congression, independent of any effects on the spindle (*Schmidt et al., 2005*). Therefore, we used the same strategy, shifting worms to the restrictive temperature for short periods and then assessing chromosome segregation. Following brief dynein inactivation using this strategy, we observed a significant fraction of anaphase spindles that had lagging chromosomes (2/12 after a 5 min shift, 15/20 after a 7.5 min shift, and 15/21 after a 10 min shift), supporting a role for dynein in chromosome segregation (*Figure 5E*).

## Chromosomes lacking rings display defects in congression and segregation

Integrating our findings, our data support the model that there are opposing forces acting on oocyte chromosomes during congression and segregation, with plus-end directed forces originating in the rings countered by accumulating minus-end forces. To further test this idea, we assessed the behavior of chromosomes lacking rings by analyzing mutants where particular sets of homologous chromosomes are unable to pair (*him-8(me4)* and *zim-2(tm574)*, where the X chromosome (*Phillips et al., 2005*) and chromosome V (*Phillips and Dernburg, 2006*) are affected, respectively), resulting in five bivalents and two unpaired chromosomes (univalents). These univalents were not recognized by antibodies against MPM-2 or the CPC component AIR-2/Aurora B, indicating that they do not form ring structures (*Figure 6A*, asterisks), and consequently each spindle had only five rings. However, the univalents were able to load kinetochore components, as BUB-1, which localizes to both the kinetochore and the ring (*Monen et al., 2005*; *Dumont et al., 2010*), coated their outer surfaces (*Figure 6A*).

Importantly, we found that univalents lacked a plus-end directed force, since in monopolar spindles both univalents were located near the pole in every spindle we quantified (40/40 spindles; 80/80

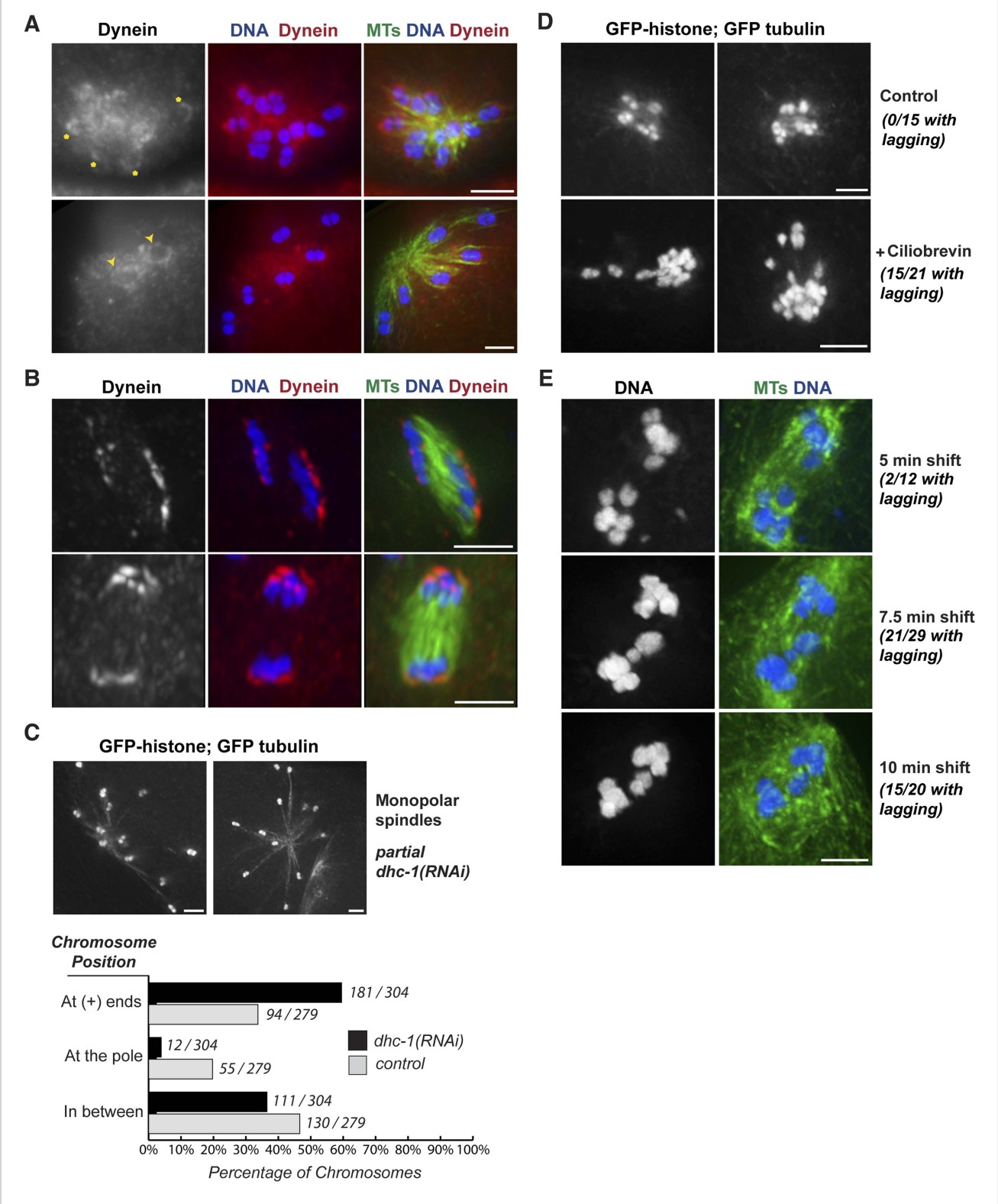

**Figure 5**. Dynein contributes to chromosome segregation on oocyte spindles. (**A**) Monopolar oocyte spindles stained for tubulin (green), DNA (blue), and DHC-1/dynein (red). The tubulin and DNA channels were deconvolved, but the dynein channel was not, to show a faint pool of dynein surrounding chromosomes. Dynein cups chromosomes, sometimes found enriched on the side facing away from the pole ('asterisks') and sometimes around the entire chromosome ('arrowheads'); partial projections are shown to highlight particular chromosomes. (**B**) Dynein localization on anaphase spindles with the

*Figure 5. continued on next page*

*Figure 5. Continued*

same colors as (**A**), except the dynein channel was deconvolved. Dynein localizes adjacent to the outside surface of separating chromosomes. (**C**) Images of monopolar spindles in a strain expressing GFP-histone; GFP-tubulin following partial dynein RNAi. Spindles are larger with chromosomes often located at the extreme ends microtubules; graph depicts chromosome positions on monopolar spindles in *dhc-1 (RNAi)* (black bars) compared to control (gray bars). (**D**) Images of monopolar anaphase spindles in a strain expressing GFP-histone; GFP-tubulin with no treatment (top row) or following incubation with 5 μM ciliobrevin for 20 min (bottom row). Following dynein inhibition, lagging chromosomes were observed in a significant fraction of spindles (15/21). (**E**) Images of bipolar anaphase spindles following dynein inhibition; DNA is shown in blue and microtubules in green. The dynein temperature-sensitive mutant *dhc-1(ct76ts)* was shifted to the restrictive temperature for 5 (top), 7.5 (middle), or 10 min (bottom); spindles with lagging chromosomes were observed under all three conditions. Bars = 2.5 μm.

The following figure supplement is available for figure 5:

**Figure supplement 1**. Strong dynein inhibition causes spindle defects.

univalents) (*Figure 6B*, bottom panel). Moreover, we found that univalents exhibited congression defects on bipolar spindles. This phenotype was especially apparent following metaphase arrest, as most bivalents are aligned in those conditions, allowing us to more clearly quantify univalent position with relation to the metaphase plate (*Figure 6B*). For quantification, we assessed whether the univalent was found in the middle region of the spindle where the bivalents were located (Zone 1), close to but not completely overlapping with the bivalents (Zone 2), or in the regions closer to the poles (Zone 3). Univalents were fairly evenly distributed between these three categories, consistent with the univalents being positioned at random on bipolar spindles.

Next, we assessed the behavior of univalents during anaphase. First, we found that univalents remained as intact units during anaphase and did not release cohesion between sister chromatids. This observation is consistent with our finding that separase relocalizes to the rings at the metaphase to anaphase transition (*Figure 2C*); since univalents lack ring components (*Figure 6A*), separase is likely not targeted to them, preventing cohesion release. Most often, we found that these intact univalents lagged during anaphase (*Figure 6C*, top row; 76%). However, 24% of univalents did not lag and instead appeared to segregate with the rest of the chromosomes (*Figure 6C*, bottom row). These findings are consistent with the interpretation that the position of the univalent when anaphase is triggered determines its behavior during segregation. Univalents that were located close to a pole prior to segregation (zone 3) likely remain close to that same pole during anaphase; minus-end forces could properly act on that univalent and therefore it would appear to segregate with the rest of the chromosomes. However, a univalent located close to the chromosomes (zones 1 and 2) would become trapped in the center, as it would be caught between overlapping bundles of microtubules, with minus-end forces acting on both sides of the univalent pulling it in opposite directions. Therefore, the behavior of univalents during congression and segregation supports a model in which a balance between opposing plus- and minus-end directed forces dictate chromosome position on the acentrosomal spindle and would represent a case in which the plus-end force is eliminated.

## Discussion

### A new model for chromosome segregation on acentrosomal spindles

Taken together, our data support a model in which chromosome segregation on the *C. elegans* acentrosomal spindle is driven by minus-end directed movement along lateral microtubule bundles (*Figure 7A*), revealing a new kinetochore-independent strategy for partitioning chromosomes. The spindle reorganizes at the metaphase to anaphase transition to enable this mode of segregation, by broadening the spindle poles and thereby creating channels that are open from pole to pole. Chromosomes move through these channels until they are located at the tips of the microtubules at the minus ends, at which point the channels close and the spindle disassembles. Coupling spindle pole broadening to anaphase onset could potentially serve a dual purpose in regulating chromosome dynamics in the context of our model. First, pole broadening would be important in anaphase, as it would serve to open a path for chromosomes to move all the way to the microtubule minus ends. Moreover, we speculate that having the poles more focused in prometaphase could serve the opposite purpose; since chromosomes move along lateral microtubule bundles during congression as

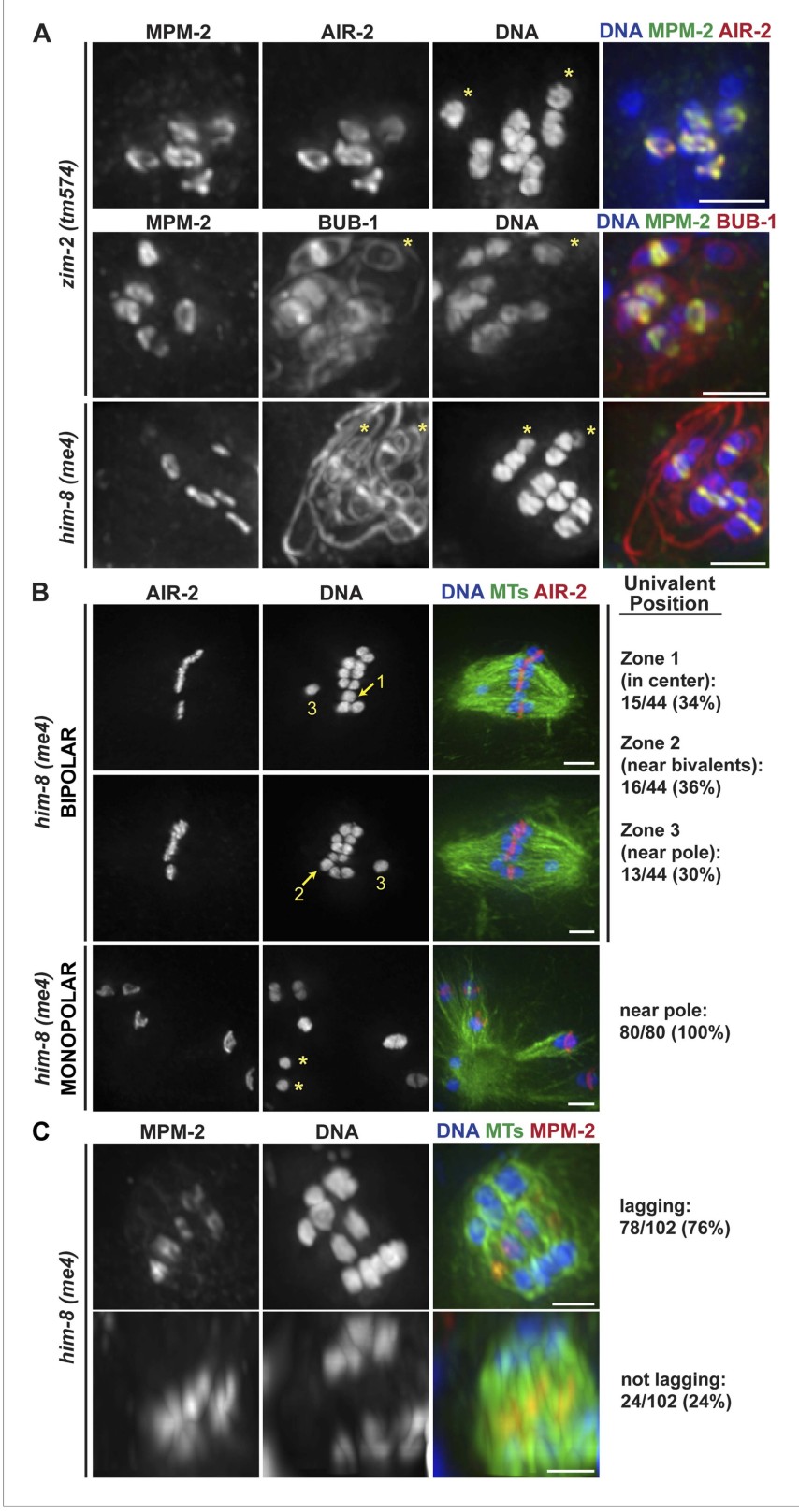

**Figure 6**. Univalents lacking rings exhibit congression and segregation errors. (**A**) Oocytes from *him-8(me4) and zim-2(tm574)* worms stained for DNA (blue), AIR-2 or BUB-1 (red), and MPM-2 (green); each of these strains has two univalents ('asterisks'). Rings are not detected on univalents, but kinetochore staining is present (BUB-1; bottom two
*Figure 6. continued on next page*

*Figure 6. Continued*

rows). (**B**) Metaphase-arrested *him-8(me4)* bipolar and monopolar spindles, stained for DNA (blue), AIR-2 (red), and tubulin (green). During congression, univalents are randomly positioned on the bipolar spindle (top two rows), but were located near the pole on every monopolar spindle we observed (bottom). For quantification of bipolar spindles, univalents were scored as being at the center of the metaphase plate (zone 1), partially overlapping with the bivalents (zone 2), or not overlapping (zone 3). Examples of each category are indicated on the images. (**C**) *him-8 (me4)* anaphase spindles stained for DNA (blue), MPM-2 (red), and tubulin (green). During segregation, univalents are either found near other segregating chromosomes (bottom) or lag behind in the center (top). Bars = 2.5 µm.

well, having the poles 'closed' at that stage could bias chromosomal movement towards the metaphase plate.

We further propose that chromosome movement on acentrosomal spindles is mediated through a balance of plus- and minus-end directed forces operating on chromosomes (***Figure 7B***). During congression, the ring provides a plus-end directed force that promotes movement to the metaphase plate and during the same period minus-end directed forces begin to accumulate on chromosomes.

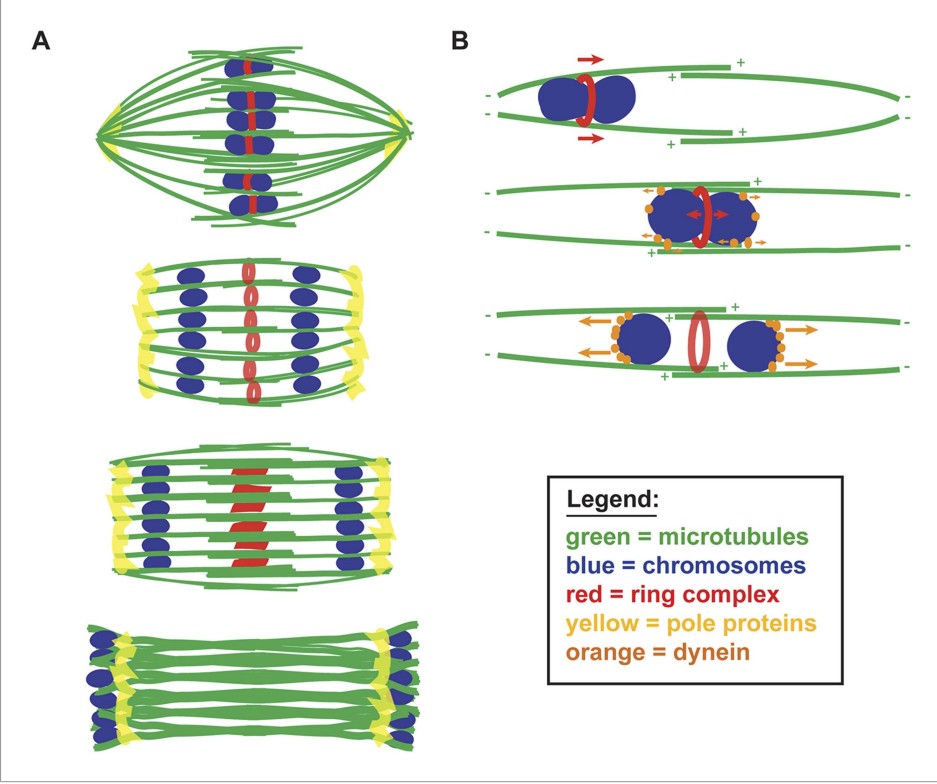

**Figure 7**. A balance of forces controls chromosome movement on the acentrosomal spindle. (**A**) Model depicting microtubules (green), chromosomes (blue), rings (red), and spindle pole proteins (yellow). During metaphase, poles are focused and chromosomes are aligned between overlapping microtubule bundles. During anaphase, poles broaden creating open channels and separated chromosomes move along lateral bundles until they are past the poles; at this stage, the channels close. In anaphase, rings are removed from chromosomes and then flatten and disassemble. (**B**) Model depicting microtubules (green), chromosomes (blue), rings (red), and dynein (orange). During congression, KLP-19 and potentially other ring components provide a plus-end directed force ('red arrows'), moving the chromosome toward the metaphase plate; during this time dynein begins to accumulate on chromosomes, resulting in weak minus-end forces ('small orange arrows'). At metaphase, the plus- and minus-end directed forces are each balanced as the chromosome associates with microtubules in an overlap zone. Ring removal at the metaphase to anaphase transition shifts this balance, triggering poleward movement. In anaphase, more dynein has accumulated on chromosomes, resulting in stronger minus-end forces ('large orange arrows').

As chromosomes then align at the metaphase plate, the plus- and minus-end directed forces would each be balanced, since at that stage the chromosomes are positioned within microtubule bundles of overlapping polarity, with both plus- and minus-ends on each side. Since our data support the idea that minus-end forces increase as metaphase progresses, we infer that these forces are likely weak early in congression (allowing plus-end forces to be dominant) and then accumulate in preparation for anaphase. Once anaphase is triggered, ring removal alters the balance of forces, enabling minus-end forces to propel separated chromosomes to the poles. Importantly, our proposed mechanism would directly couple key essential events at the metaphase to anaphase transition, as cleavage of cohesin by separase would also facilitate ring removal from chromosomes, thereby also removing a plus-end directed force. Consequently, the outcome would be coordinated poleward movement, driving chromosome segregation.

A previous study proposed a different model for kinetochore-independent segregation, where the metaphase spindle disassembles from the poles and then a new microtubule array assembles between separating chromosomes, stimulated by ring component CLASP (CLS-2). In this model, a force provided by polymerizing microtubules pushing against the inside surface of separating chromosomes is the primary candidate driving segregation (*Dumont et al., 2010*). Our findings are not consistent with key features of this model, since spindle poles do not appear to disassemble and polymerization of a new microtubule array coupled to the inside surfaces of separating chromosomes would be predicted to close the microtubule channels that we observe. Moreover, the previous model also does not account for the behavior of chromosomes during monopolar anaphase, where separated chromosomes do not move apart, pushed from the center, but instead move together to the same pole, leaving the ring behind. However, the major length changes that occur in the *C. elegans* oocyte spindle (with it shrinking and then elongating as anaphase progresses) make it likely that spindle elongation also contributes to segregation, potentially driven by CLASP-dependent microtubule polymerization as this previous study proposed. This process may be analogous to Anaphase B in mitotically dividing cells (where spindle pole separation drives sets of separating chromosomes further apart [*Walczak and Heald, 2008*]) but it would have to occur within the context of the unique spindle organization that we have now defined, with lateral microtubule bundles extending alongside separating chromosome pairs. However, even if spindle elongation contributes to segregation, our studies have now revealed another mechanism that operates in parallel, with forces operating on the chromosomes themselves driving poleward movement. This segregation mechanism could be viewed as analogous to Anaphase A, but it would be mechanistically distinct from what occurs in mitotically dividing cells, as the chromosome to pole movement that we have described does not require kinetochores and is instead mediated by motor-driven movement along lateral microtubule bundles.

Our work raises a number of intriguing questions that remain to be addressed. First, our current data do not completely explain how chromosomes escape the zone of overlapping microtubules in the center of the spindle to ensure that they move to the correct poles. Although dynein is asymmetrically localized on separating chromosomes (as we do not detect it on the inside surface), it is still possible that some motors could associate with microtubule bundles oriented towards the incorrect pole (*Figure 7B*). Therefore, we speculate that an additional mechanism may exist to give the separating chromosomes an initial push apart; this mechanism could involve a morphological change in the ring structure as it is removed (a constriction or elongation), a small amount of CLASP-dependent microtubule polymerization, or a currently undiscovered mechanism. It is also possible that the rings could simply serve as physical barriers between separating chromosomes, blocking the center of the channel and therefore preventing chromosomes from moving in the wrong direction, giving motors associated with properly oriented microtubule bundles an advantage. Another related question that remains to be addressed is the polarity of microtubules within the bundles and the exact length of the overlap zone, which would rely on careful analysis by electron microscopy. We favor the idea that microtubules within the laterally associated bundles are organized primarily, if not completely, with the polarity depicted in *Figure 7*, as bundles organized with a defined polarity would best explain the opposing forces we observe acting on rings and chromosomes on monopolar spindles, and the phenotype of dynein depletion on these spindles (where chromosomes are then positioned closer to the outside of the aster). However, our studies do not rule out the possibility that the spindle may also contain non-bundle associated microtubules of mixed polarity that could affect chromosome behavior.

## A role for dynein in chromosome dynamics on the acentrosomal spindle

Our studies have also revealed a role for dynein in mediating chromosome dynamics on the acentrosomal spindle. First, we found that dynein exhibits chromosomal localization. In prometaphase and metaphase bipolar spindles, chromosomal dynein is faint and often difficult to distinguish above the spindle-localized pool, suggesting that there is only a small amount of the motor on the chromosomes. However, anaphase localization is more clearly apparent, supporting the idea that dynein accumulates on chromosomes in preparation for driving anaphase movements. Pre-anaphase chromosomal dynein localization can be more clearly visualized on monopolar spindles, where the motor is often seen more concentrated on the end of the chromosome facing away from the pole. We speculate that this asymmetric pattern could reflect the mechanism by which dynein loads onto the chromosomes, for example, if it walks towards the chromosomal surface from the microtubule plus ends. This loading mechanism could possibly facilitate accumulation of the motor on the outer surfaces of chromosomes prior to anaphase, since when chromosomes are at the metaphase plate with overlapping plus ends nearby, dynein loading via this mechanism might be especially efficient, allowing the motor to accumulate on chromosomes in preparation for anaphase initiation.

We also observed chromosome segregation defects following dynein inhibition, with lagging chromosomes in a significant fraction of both monopolar and bipolar anaphase spindles. However, these phenotypes were not completely penetrant, as some chromosomes did exhibit poleward movement in each of these cases. One possibility is that this movement is due to residual dynein activity; since dynein also plays a role in spindle assembly, we had to use partial inactivation or a fast-acting temperature-sensitive mutant and each of these strategies could potentially leave a significant pool of active dynein. A second possibility is that there are additional factors contributing to the minus-end force operating on chromosomes (for example, poleward flux or other minus-end directed motors). This second possibility could also apply to the plus-end directed force; we previously demonstrated that the kinesin motor KLP-19 in the ring provides a polar ejection force using our monopolar spindle assay (*Wignall and Villeneuve, 2009*), but it is possible that other factors operate in parallel to promote congression. However, even if there are multiple components of either the plus- or minus-end directed forces acting on chromosomes, the mechanism that we propose for segregation, with a hand-off between these forces, could still operate as proposed (*Figure 7B*).

Another interesting observation that emerged from our studies is that partial dynein inhibition results in especially large monopolar spindles (*Figure 5C*, the spindles shown are approximately 50% larger than normal monopolar spindles), suggesting that dynein also regulates spindle length. A full understanding of this phenotype will require further experimentation, though our observation is consistent with work in other systems, where dynein depletion has also been reported to cause larger spindles (*Gaetz and Kapoor, 2004*; *Morales-Mulia and Scholey, 2005*). However, relevant to this study, larger monopolar spindles were particularly useful in assessing the contribution of opposing forces on the chromosomes; with longer microtubules, the positioning of chromosomes at the extreme plus ends was particularly clear, providing support for the idea of a minus-end directed force provided by dynein acting on the chromosomes.

## A conserved role for lateral microtubule–chromosome associations in mediating chromosome dynamics

In conclusion, our work defines a new mechanism of kinetochore-independent chromosome segregation and highlights a role for lateral microtubule–chromosome associations in driving chromosome movements. Notably, lateral interactions have been previously described in mitotically dividing cells and have been shown to play a role in chromosome congression. For example, kinetochores associate laterally with microtubules before forming end-on attachments (*Tanaka, 2012*), a class of kinesins on chromosome arms promotes congression (*Vanneste et al., 2011*), and chromosomes can congress under experimental conditions where kinetochore-fiber formation is inhibited (*Cai et al., 2009*). However, our work now demonstrates that these associations can be the primary drivers of chromosome movement in an unperturbed system, and that in addition to congression, they can also provide the force to power chromosome segregation.

Interestingly, our findings also have parallels with previous work on the mammalian oocyte spindle. First, it has been shown that acentrosomal spindles in mouse oocytes don't have end-on kinetochore microtubule-attachments prior to achieving metaphase alignment (*Brunet et al., 1999*), and fluorescence

imaging of prometaphase is suggestive of the presence of lateral microtubule bundles surrounding the chromosomes (with a lower density of microtubules at the ends) (*Schuh and Ellenberg, 2007*). While kinetochore-microtubule attachments are established after metaphase alignment (*Brunet et al., 1999*; *Kitajima et al., 2011*) and have been demonstrated during anaphase (*FitzHarris, 2012*), there is also evidence for kinetochore-independent mechanisms. An intriguing study reported that DNA-coated beads injected into mouse oocytes formed acentrosomal spindles and that when anaphase was induced, these beads exhibited dynein-dependent poleward movements (*Deng et al., 2009*). Therefore, we suggest that the mechanism that we describe could also operate in mammalian oocytes during anaphase. Although the primary driver of chromosome-to-pole movement in these cells may be the canonical kinetochore-driven mechanism, lateral microtubule associations established to promote congression could remain intact in anaphase, and dynein localized on the chromosomes could provide a force to help move chromosomes along these microtubules towards minus ends. Therefore, by studying anaphase in *C. elegans* oocytes, we have revealed a new mechanism driving chromosome segregation on acentrosomal spindles that may be widely applicable to understanding chromosome behavior during oocyte meiosis.

## Materials and methods

### Strains

'Wild type' refers to N2 (Bristol) worms, and 'control' refers to RNAi vector control, temperature-sensitive strains grown at the permissive temperature, or worms not treated with inhibitor. A chart showing all strains used is shown in the Supplementary material (*Supplementary file 1*).

### RNAi

From a feeding library (*Fraser et al., 2000*; *Kamath et al., 2003*), individual RNAi clones were picked and grown overnight at 37°C in LB with 100 µg/ml ampicillin. Saturated cultures were spun down, plated on NGM (nematode growth media) plates supplemented with 100 µg/ml ampicillin and 1 mM IPTG (isopropyl-β-d-thiogalactopyranoside), and dried overnight. L1 worms (synchronized by bleaching gravid adults and hatching overnight without food) were then plated on RNAi plates and grown to adulthood at 15° for 5 days. For partial *dhc-1(RNAi)*, worms were grown until the L4 stage on regular NGM/OP50 plates and then transferred to the RNAi plate 24–48 hr before imaging.

### Metaphase arrest and dynein inhibition

Metaphase arrest was achieved by either RNAi-mediated depletion of *emb-30* or by shifting *emb-27 (g48)* worms to 25°C for 4–5 hr. Dynein was inhibited by *dhc-1* partial RNAi (described above), by shifting *dhc-1(ct76)* worms to 26°C for 5–15 min, or by soaking worms for 20 min in 5 µM ciliobrevin (HPI-4, Sigma, St. Louis, MO).

### Immunofluorescence and antibodies

Immunofluorescence was performed as previously described (*Oegema et al., 2001*), with slides fixed in methanol for 40–45 min. The following antibodies were used: rabbit anti-AIR-2 (1:500; gift from Jill Schumacher), rabbit anti-BUB-1 (1:2800; gift from A Desai), rabbit anti- DHC-1 (1:100, gift from P Gönczy), rabbit anti-GFP (1:500; AbCam, Cambridge, UK), mouse anti-GFP (1:500; AbCam), rabbit anti-KLP-18 (1:10,000, gift of O Bossinger), rabbit anti-KNL-3 (1:3800; gift from A Desai), mouse anti-MPM-2 (1:500; AbCam and Millipore, Billerica, MA), rabbit anti-SEP-1 (1:200; gift from A Golden), mouse anti-α-tubulin-FITC (1:500; Sigma). Alexa-flour-conjugated secondary antibodies (Invitrogen, Carlsbad, CA) were used at 1:500.

### Microscopy

For most imaging, a Deltavision deconvolution microscope with a 100× objective was used (Applied Precision, part of GE Healthcare, Piscataway, NJ). This microscope is housed in the Northwestern Biological Imaging Facility supported by the NU Office for Research. Image stacks were obtained at 0.2 µm z-steps for fixed samples and deconvolved using SoftWoRx (Applied Precision). For images denoted as super-resolution, a Deltavision OMX 3D-SIM with an Olympus (Shinjuku, Tokyo, Japan) 100× UPlanSApo 1.4 NA objective was used (Applied Precision), located in the Light Microscopy Imaging Center (Indiana University). Images were captured at 0.125 µm z-steps and processed using

SoftWoRx (Applied Precision) and IMARIS 3D imaging software (Bitplane, Zurich, Switzerland). Control and RNAi worms were processed in parallel, and image exposure times were kept constant within each experiment.

## Time-lapse imaging

For live imaging, GFP- or mCherry-expressing worms were picked into a solution of tricaine (2%) and tetramisole (0.4%), and incubated for 40 min. Worms were then pipetted onto a 3% agar pad, covered with a coverslip, and imaged immediately. Image stacks were obtained at 1 μm z-steps at 10-s intervals using 2 × 2 binning, and then deconvolved. Video images are full projections of data stacks encompassing the entire meiotic spindle.

## Image analysis and quantification (for specific figures)

### Figure 4

In Imaris 'Surpass' view, 3D images were rotated to allow assessment of bivalent location on the monopolar spindle. In addition, ring morphology and position relative to the bivalent was assessed, using the volume rotations. In 10 'chromosome in' images and 7 'chromosome out' images, we saw no ring stain. We attribute this to lack of antibody staining, so these were not included in the numbers presented in the figure. In 3 images we observed 'bivalents in' with unstretched rings stretched on the bivalent, we classify this as spindle formation/early prometaphase on the monopolar spindle.

### Figure 6

To assess the position of univalents relative to the bivalents, each 3D image was rotated in 'Surpass' view to allow for a view showing aligned bivalents (using midbivalent ring location as a guide). If a univalent was aligned with the midbivalent ring, it was categorized as 'zone 1'. If a univalent was partially in the plane of the bivalents but not completely overlapping, it was categorized as 'zone 2'. If it was outside the plane of the bivalent, it was categorized as 'zone 3'. For anaphase assessment, 'lagging' univalent refers to a univalent positioned closer to the midzone of the anaphase spindle than the segregating chromosomes.

### Videos 2–5

A super-resolution 3D image was visualized in the animation window. Rotation positions were specified as frames within the video. Orthogonal Slice view was used to look at individual slice views through the 3D spindle and specified as frames within the animation. The animation was then recorded in Imaris using a Quicktime format.

## Acknowledgements

We thank members of the Wignall lab for support and discussions, Richard Carthew, Amanda Davis, Rebecca Heald, Carissa Heath, Timothy Mullen, and Ian Wolff for critical reading of the manuscript and Olaf Bossinger, Bruce Bowerman, Arshad Desai, Andy Golden, Pierre Gönczy, Frank McNally, Jill Schumacher, and Susan Strome for reagents. Additionally, some strains were provided by the CGC, which is funded by the NIH Office of Research Infrastructure Programs (P40 OD010440). This work was supported by the Chicago Biomedical Consortium with support from the Searle Funds at The Chicago Community Trust, by an American Cancer Society Internal Research Grant (ACS-IRG 93-037-18), by a Basil O'Connor Award from the March of Dimes, by a V Scholar Award from the V Foundation for Cancer Research, and by a Damon Runyon-Rachleff Innovation Award (all to SMW).

## Additional information

### Funding

| Funder | Grant reference | Author |
|---|---|---|
| March of Dimes Foundation | Basil O'Connor Starter Scholar Award | Sarah M Wignall |
| Damon Runyon Cancer Research Foundation | Runyon-Rachleff Innovation Award | Sarah M Wignall |

| Funder | Grant reference | Author |
|---|---|---|
| V Foundation for Cancer Research | V Scholar Award | Sarah M Wignall |
| Chicago Community Trust | Chicago Biomedical Consortium Recruitment Award | Sarah M Wignall |
| American Cancer Society | Internal Research Grant ACS-IRG 93-037-18 | Sarah M Wignall |

The funders had no role in study design, data collection and interpretation, or the decision to submit the work for publication.

#### Author contributions

CCM, KMT-S, Conception and design, Acquisition of data, Analysis and interpretation of data, Drafting or revising the article; MVT, JAP, Acquisition of data, Analysis and interpretation of data; SMW, Conception and design, Analysis and interpretation of data, Drafting or revising the article

## Additional files

#### Supplementary file

• Supplementary file 1. Complete list of strains used in this study.

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
