## [Decision Letter]

Thank you for sending your work entitled “Kinetochore-independent chromosome segregation driven by lateral microtubule bundles” for consideration at *eLife*. Your article has been favorably evaluated by Tony Hunter (Senior Editor) and three reviewers, one of whom is a member of our Board of Reviewing Editors.

The Reviewing Editor and the other reviewers discussed their comments before we reached this decision. As you can see, all reviewers are positive and require only changes in writing and some interpretation.

The only experimental point that seems important to us is the following from Reviewer 3. We think quantifying the fluorescence would help with the interpretation of your results, and encourage you to do so.

The authors state that “chromosomes associate with lateral bundles of microtubules during anaphase” and that this is “inconsistent with a model in which a new array of microtubules polymerizes between separating chromosomes”. This is initially based on the observation that open “channels” exist in the MT spindle network between separating chromosomes in early-late anaphase. However, a pushing force could still come from polymerizing MTs within the lateral MT bundles. Furthermore, the fact that chromosomes move beyond the pole region at the end of anaphase suggests that a pushing force from within the midzone is involved. A simple quantification of MT levels (total fluorescence of the entire spindle) during different stages of meiosis would help address the issue of whether more MTs are generated during anaphase. This would help to verify the authors' claims that the spindle does not disassemble and reassemble MTs, rather, it reorganizes existing MTs during the transition from metaphase to anaphase.

*Reviewer #1*:

Canonical chromosome segregation is dependent on the kinetochore, complex of structural and regulatory proteins that are normally required for attachments between centromeres on chromosomes and the ends of microtubules. Despite their importance for normal mitotic chromosome segregation in most organsims, kinetochores are not required for chromosome segregation in acentrosomal spindles in *C. elegans* oocytes. The mechanism of this segregation is not clear. Muscat el al. shed light on this mechanism using conventional and super-resolution microscopy. During chromosome congression, a ring of proteins including the motor *klp-19* and the chromosomal passenger complex is present around the mid-region of each chromsomome pair. At anaphase, the microtubule array reorganizes to create channels where chromosomes pass through, while the rings around the mid-region are lost. The authors present a model where chromosome congression is dependent on a balance of opposing plus end and minus end directed forces. At anaphase, *klp-19* and potentially other plus end directed motors are lost, favoring minus end directed movement of chromosomes through the channels. Minus end directed movement is dependent dynein.

This is an interesting study that provides a different mechanism of kinetochore-independent segregation to that proposed in a previous study (Dumont et al., 2002) where anaphase separation of chromosomes was proposed to occur via assembly of a new microtubule array that would push chromosomes apart. The channels that the authors observe in this study make this scenario less likely, and instead support a different model where chromosomes move along a reorganized parallel microtubule array by means of lateral associations.

The experiments are well designed, the imaging is of high quality, and the conclusions are well supported. I recommend publication and have only a couple of minor suggestions, both pertaining to the “balance of forces” model presented in Figure 7.

1) I'm confused why KLP-19 is not included in the model, while dynein is. Though it is part of the “ring” complex, it would be helpful if this were shown, or at least mentioned in the figure legend.

2) A color coded legend would make the model much easier to interpret.

*Reviewer #2*:

In this paper the authors have analyzed acentrosomal chromosome segregation in *C. elegans* female meiosis. The authors present evidence that partitioning of sister chromosomes/chromatids is driven by opposing forces: 1. plus-end directed forces as exerted by a ring-shaped protein complex around the chromosomes, and 2. minus-end directed forces as caused by the action of dynein. The authors have presented a model in which removal of the central ring is accompanied with an opening of the spindle poles. Both the spindle pole opening and the removal of the ring-shaped complex are supposed to trigger poleward movement of chromosomes/chromatids along bundles of microtubules.

The authors have presented clear data towards understanding the largely unknown field of female meiosis in *C. elegans*. In accord with previous observations, the authors have presented clear evidence that the segregation process in female meiosis is kinetochore-independent.

From the data presented, however, it is not clear how the meiotic spindle in *C. elegans* is organized. Although not within the scope of this light microcopy study, I would like the authors to comment on the following issues related to spindle architecture:

1) The presented model is based on the idea that chromosomes/chromatids move laterally along microtubule bundles. What about the polarity of the microtubules in the system? Isn't it a requirement of the model that all microtubules are oriented in a way that the minus ends of the microtubules are facing towards the acentrosomal poles?

2) The model does also imply that the number of microtubules from metaphase to early anaphase is not changing? The authors have presented a model in which an elongation of the microtubules is the major aspect of spindle elongation. This is in contrast to the Dumont model of a regrowth of microtubules between the segregating chromosomes/chromatids. Are the Dumont and Muscat models mutually exclusive?

It is clear that light microscopy cannot fully answer such important spindle-related questions and it is certainly beyond the scope of this manuscript to do additional EM experiments. Despite the lack of ultrastructural information in this manuscript, however, this is a valuable contribution to the current discussion on *C. elegans* meiosis. In general, I recommend publication of the manuscript in *eLife*. However, I would like the authors to comment on the questions raised, before this manuscript can be finally accepted.

*Reviewer #3*:

In 2010, the Desai lab showed that kinetochores were not required for the segregation of chromosomes during female meiosis in *C. elegans*, but that CLASP-dependent MT polymerization was required. In their model, MT polymerization between separated chromosomes provides a pushing force for chromosome segregation. In the submitted article, Muscat et al. present a new model for chromosome segregation in female meiosis. Using high-resolution imaging, they show that MTs are not present directly between the separating chromosomes in early anaphase. Instead, lateral microtubule bundles contact the sides of chromosomes throughout the various stages of the meiotic cell division cycle. The authors experimentally determine that MT motor KLP-19 provides a plus-end directed activity for chromatid congression, and dynein provides an opposing minus-end force that is utilized for anaphase chromosome movement along the same MT bundles. The data presented is solid and the model is supported by most of the observations. I have a few concerns and suggestions, but overall, this work provides a reasonable model for chromosome segregation in acentrosomal spindles, and warrants publication in *eLife*.

Suggested experiments/presentation of data:

1) The authors state that “chromosomes associate with lateral bundles of microtubules during anaphase” and that this is “inconsistent with a model in which a new array of microtubules polymerizes between separating chromosomes”. This is initially based on the observation that open “channels” exist in the MT spindle network between separating chromosomes in early-late anaphase. However, a pushing force could still come from polymerizing MTs within the lateral MT bundles. Furthermore, the fact that chromosomes move beyond the pole region at the end of anaphase suggests that a pushing force from within the midzone is involved. A simple quantification of MT levels (total fluorescence of the entire spindle) during different stages of meiosis would help address the issue of whether more MTs are generated during anaphase. This would help to verify the authors' claims that the spindle does not disassemble and reassemble MTs, rather, it reorganizes existing MTs during the transition from metaphase to anaphase.

Other points:

1) It looks like MTs do not form bundles in the monopolar spindles. If bundles do not form, is it fair to compare the movement of chromosomes in this system to a bipolar array?

2) In Figure 3, it is difficult to determine whether the chromosomes are moving along the sides of microtubules in these panels (and the supplemental movie) or whether they are moving in conjunction with MT depolymerization. Could MT depolymerization be the main driver of minus-end movement?

3) Some aspects of the model are unclear. Muscat et al. suggest that MT polymerization is not required for chromosome segregation, so how do the authors reconcile data from the Desai lab that showed that loss of CLASP abolished segregation? Could polymerization along the lateral bundles contribute to the segregation, or do the authors think that CLASP is doing something other than promoting MT polymerization?

The authors suggest that, during congression, the plus-end forces are active but that the minus-end forces start to accumulate (Discussion, second paragraph). But in the next sentence, it is the overlapping bundles of MTs with mixed polarity that causes the movement of chromosomes to stop in the middle of the spindle.

First, how can the congression work if dynein is also present? Figure 7 suggests that plus-end forces are active during congression, so is dynein accumulating but not active until after congression? What is a “weak” minus-end force, and what changes to make it stronger?

Second, it is not obvious how the chromosomes escape the overlapping MT zone as depicted in the last two panels of Figure 7 (ring removal alone cannot explain it, unless the overlapping MT zone becomes extremely small). Is it possible that something else (e.g. CLASP-dependent MT polymerization?) is required to push the chromosomes beyond the region of mixed MT-polarity, to allow the minus-end forces to take over? Or could CLASP simply provide many “loose”, non-spindle-pole MTs that reduce the opposing plus-end resistance simply by competition? Or is it that MT plus-end directed forces acting between anti-parallel MTs within the bundles eventually shrinks the overlap region to the size depicted in the cartoon? Please provide some suggestion(s) for how this transition might occur.

---

## [Author Response]

Reviewer #1:

*1) I'm confused why KLP-19 is not included in the model, while dynein is. Though it is part of the “ring” complex, it would be helpful if this were shown, or at least mentioned in the figure legend*.

We originally did not label KLP-19 in our diagram because we did not want to give the incorrect impression that the ring was comprised only of KLP-19 or that we had evidence that KLP-19 is the only plus-end directed force acting on chromosomes. However, we have taken the reviewer’s helpful suggestion and have now updated the figure legend to include mention of KLP-19, which avoids this problem.

*2) A color coded legend would make the model much easier to interpret*.

We have now added a legend.

Reviewer #2:

*From the data presented, however, it is not clear how the meiotic spindle in* C. elegans *is organized. Although not within the scope of this light microcopy study, I would like the authors to comment on the following issues related to spindle architecture*:

*1) The presented model is based on the idea that chromosomes/chromatids move laterally along microtubule bundles. What about the polarity of the microtubules in the system? Isn't it a requirement of the model that all microtubules are oriented in a way that the minus ends of the microtubules are facing towards the acentrosomal poles*?

We do not have direct evidence by EM that microtubules within the bundles are organized in the way we depict, and we have now acknowledged this point and brought up the issue of microtubule polarity in the Discussion. As the reviewer suggests, we favor the idea that microtubules in the laterally-associated bundles are organized primarily (if not completely) with the polarity depicted in Figure 7; there may also be non-bundle associated microtubules of mixed polarity within the spindle as well, but our model focuses only on bundle-associated microtubules. Although our experiments don’t definitively show the polarity of microtubules within the bundles, we think that this interpretation is most consistent with our data. In monopolar spindles (where it is easier to distinguish between poleward and anti-poleward forces because there is no zone where bundles from opposing poles overlap), rings consistently stretch in one direction during metaphase arrest and chromosomes are subjected to an opposite force (Figure 4). The simplest explanation for this behavior is that microtubules within the bundles are all oriented the same way; if microtubules were of mixed polarity, we would predict that the rings would not display such consistent behavior (always stretching in the same direction). Since we have identified a plus-end directed force (KLP-19) in the ring, and dynein inhibition causes chromosomes to be found further from the central pole, our data are most consistent with an organization in which the minus ends are in the center of the aster and the plus ends extend outwards. In the bipolar spindle, this would mean that the minus ends are focused at the two poles. Although we think that performing a detailed analysis of anaphase spindle morphology by EM is a worthy thing to do and is an important test of our model, we agree with the reviewer’s later comment that it is beyond the scope of this study.

*2) The model does also imply that the number of microtubules from metaphase to early anaphase is not changing? The authors have presented a model in which an elongation of the microtubules is the major aspect of spindle elongation. This is in contrast to the Dumont model of a regrowth of microtubules between the segregating chromosomes/chromatids. Are the Dumont and Muscat models mutually exclusive*?

Some aspects of our model and the Dumont model are mutually exclusive and some are not; we have now added a section to the Discussion discussing the previous model in more detail to try to clarify these points. One discrepancy relates to the idea presented in the Dumont model that the metaphase spindle disassembles from the poles so that a new microtubule array can form between separating chromosomes; this is not consistent with our observations that microtubules extend alongside or beyond chromosomes throughout anaphase and that pole proteins also remain associated with spindles, suggesting that the poles don’t disassemble. Another major difference relates to the proposed force separating chromosomes in the two models. Dumont, et al. suggest that the most likely candidate for the force driving chromosome segregation is microtubule polymerization between the separating chromosomes; microtubules coupled to the inside surfaces of separating chromosomes would serve to push them apart. This idea is not consistent with our observation of open channels within the spindles (new polymerization associated with the inside surfaces of chromosomes would serve to close the channels). Therefore, some aspects of the two models are mutually exclusive. However, the idea put forward by Dumont, et al. that microtubule polymerization contributes to segregation is certainly possible and we think quite likely; the spindle elongates significantly during anaphase and we favor the idea that microtubule polymerization contributes to this process. Our data do not directly address the mechanisms promoting spindle elongation or whether microtubule polymerization is required for segregation, as our assays are designed to focus on the forces acting on the chromosomes themselves. Therefore, we do not know whether microtubule number within the spindle changes from metaphase to anaphase and then subsequently throughout anaphase progression. We expect that future studies addressing this issue will result in a more complete understanding of how these two aspects of anaphase spindle behavior (chromosome to pole movement along lateral bundles, reported here, and spindle elongation) together facilitate segregation.

Reviewer #3:

*Suggested experiments/presentation of data*:

*1) The authors state that “chromosomes associate with lateral bundles of microtubules during anaphase” and that this is “inconsistent with a model in which a new array of microtubules polymerizes between separating chromosomes”. This is initially based on the observation that open “channels” exist in the MT spindle network between separating chromosomes in early-late anaphase. However, a pushing force could still come from polymerizing MTs within the lateral MT bundles. Furthermore, the fact that chromosomes move beyond the pole region at the end of anaphase suggests that a pushing force from within the midzone is involved. A simple quantification of MT levels (total fluorescence of the entire spindle) during different stages of meiosis would help address the issue of whether more MTs are generated during anaphase. This would help to verify the authors' claims that the spindle does not disassemble and reassemble MTs, rather, it reorganizes existing MTs during the transition from metaphase to anaphase*.

We did not intend to give the impression that we think that microtubule polymerization does not contribute to segregation on bipolar spindles; our experiments neither support nor rule out this idea. As discussed above in response to reviewer 2, the idea that microtubule polymerization may contribute to chromosome separation during the process of spindle elongation is entirely compatible with our model (e.g., by elongation of the microtubule bundles), and microtubule polymerization in this context could be promoted by CLASP, as Dumont et al. suggest. We had tried to account for this possibility in the original manuscript with the sentence: “However, the major length changes that occur in the *C. elegans* oocyte spindle (with it shrinking and then elongating as anaphase progresses) make it likely that spindle elongation also contributes to segregation, potentially driven by microtubule polymerization as this previous study proposed.”

However, we have now tried to more clearly articulate this point, with an expanded discussion of the differences between the two models and which features are compatible/incompatible, which will hopefully prevent further confusion. We have also revised the quoted sentence to read “…are inconsistent with a model in which a new array of microtubules polymerized between separating chromosomes provides a pushing force on their inside surfaces to drive separation”, to be more precise about the feature of the previous model that is most incompatible with our data.

In addition to these changes to the text, we also took the reviewer’s suggestion and attempted to quantify our fluorescence images, to see if we could find evidence to either support or disprove the idea of microtubule polymerization occurring during anaphase. We did a number of new imaging experiments, where we fixed large numbers of embryos and collected as many anaphase images as we could per experiment, using uniform parameters (e.g., exposure, image size, images from an individual experiment taken on the same day, etc.). We then processed each undeconvolved image using the analysis software Imaris v 8.0.2. We used the Surfaces module to calculate the size of the surface representing the spindle, keeping parameters applied to each image the same, and we then recorded the sum intensity of tubulin fluorescence from each surface. For each image we also calculated chromosome distance by generating a surface model for each cluster of separating chromosomes, allowing the software to call the center of each surface, and measuring the distance between the two center points. Using this strategy, we obtained the following intensity values for a set of anaphase spindles from one of our experiments, arranged from shortest chromosome distance (i.e., early anaphase) to largest (late anaphase):CHROMOSOME DISTANCE (UM)SUM INTENSITY2.005.9E+072.016.7E+072.358.2E+062.614.7E+072.626.5E+073.105.4E+073.244.5E+073.306.1E+063.313.5E+063.412.9E+073.596.5E+073.999.3E+063.995.9E+064.169.0E+064.501.9E+065.082.4E+075.942.1E+07

From this data, we did not see a consistent trend that would allow us to argue either for or against microtubule polymerization occurring during anaphase. We think that the variations we observe are more likely the result of experimental variation (e.g., fixation differences that lead to differences in the efficiency of staining) rather than reflecting actual changes in microtubule number. We tried extensively to optimize this procedure, imaging both metaphase (not shown) and anaphase spindles, but we saw a similar level of variation in each experiment. We also used a number of different parameters in Imaris to calculate spindle volume, in case that was the source of variability, and we also performed a set of calculations where we manually set a region of interest around each spindle by eye, to rule out the possibility that the program was not accurately calculating spindle volume. However, we observed a similar level of variation no matter what method we used. We therefore think that definitively answering this question will not be possible using fixed imaging, and would be better determined by high resolution live imaging, where timepoints within the same spindle can be compared. Because we have not yet optimized live imaging conditions at a resolution that would be sufficient to accurately calculate microtubule density in the entire spindle volume, and also because neither outcome of this experiment would negatively impact our model (as our model focuses more on the forces acting on chromosomes and not on the molecular mechanisms that mediate spindle reorganization) we request that this experiment be viewed as the subject of future investigation, beyond the scope of this study.

*Other points*:

*1) It looks like MTs do not form bundles in the monopolar spindles. If bundles do not form, is it fair to compare the movement of chromosomes in this system to a bipolar array*?

In a previous study, we demonstrated that microtubules in monopolar spindles do form bundles that form lateral associations with bivalents, just as they do in bipolar spindles. This data is reported in [46], Nature Cell Biology. Bundles are shown in Figure 2 (monopolar spindles) and Figure 2 (bipolar). These bundles are apparent in images that are partial projections of z-stacks immediately surrounding individual bivalents; in full projections other microtubules in the spindle often obscure the bundles (which is the case in most of our images in the current manuscript). Therefore, although we did not present our images in a way that highlights this particular feature of monopolar spindle organization, we think that the presence of lateral bundles surrounding bivalents in both monopolar and bipolar spindles makes it possible to compare the movement of chromosomes in these two contexts. To make this point more clear to readers, we have added a sentence to the text and cited our previous work.

*2) In*
Figure 3*, it is difficult to determine whether the chromosomes are moving along the sides of microtubules in these panels (and the supplemental movie) or whether they are moving in conjunction with MT depolymerization. Could MT depolymerization be the main driver of minus-end movement*?

We were also concerned about this possibility. However, we think that two lines of evidence support the interpretation that microtubule depolymerization is not the main driver of poleward movement. First, spindles are much larger following MCAK depletion (with clear microtubules extending beyond the chromosomes in anaphase) and separated chromosomes still move to the center of the monopolar spindle (Figure 3). Second, during metaphase arrest we observe a significant number of monopolar spindles where chromosomes exhibit anaphase-like poleward movement after ring removal (Figure 4). However, these monopolar spindles remain large since they are still in metaphase, making it unlikely that spindle shrinkage and microtubule depolymerization drives these poleward movements.

*3) Some aspects of the model are unclear. Muscat et al. suggest that MT polymerization is not required for chromosome segregation, so how do the authors reconcile data from the Desai lab that showed that loss of CLASP abolished segregation? Could polymerization along the lateral bundles contribute to the segregation, or do the authors think that CLASP is doing something other than promoting MT polymerization*?

This issue is addressed in response to major point #1, above.

*The authors suggest that, during congression, the plus-end forces are active but that the minus-end forces start to accumulate (Discussion, second paragraph). But in the next sentence, it is the overlapping bundles of MTs with mixed polarity that causes the movement of chromosomes to stop in the middle of the spindle*.

*First, how can the congression work if dynein is also present?*
Figure 7
*suggests that plus-end forces are active during congression, so is dynein accumulating but not active until after congression? What is a “weak” minus-end force, and what changes to make it stronger*?

We have expanded our description of our dynein data in the manuscript to clarify these questions. We have done extensive imaging of dynein during meiosis using a variety of reagents/antibodies, and our results are consistent with the idea that dynein is present at low levels on chromosomes during prometaphase/metaphase; we can see faint dynein cupping chromosomes but it is usually difficult to distinguish from spindle-localized dynein. The images we show of dynein localization on monopolar spindles (Figure 5) are clearer (probably because the chromosomes are no longer located in a dense zone of overlapping microtubules that would obscure the chromosomal pool), but the staining is still faint and the images we show are not deconvolved, which was necessary to preserve the signal. However, it is much easier to visualize chromosomal dynein in anaphase, even in deconvolved images (Figure 5), suggesting that the amount of dynein on the chromosomes increases prior to or during anaphase. We therefore suggest that the amount of dynein on the chromosomes contributes to the strength of the force. Early in congression the minus-end force would be weaker and the plus-end forces would be dominant, but then dynein would accumulate to help drive segregation. This idea is now more clearly articulated in the text and depicted in Figure 7.

*Second, it is not obvious how the chromosomes escape the overlapping MT zone as depicted in the last two panels of*
Figure 7
*(ring removal alone cannot explain it, unless the overlapping MT zone becomes extremely small). Is it possible that something else (e.g. CLASP-dependent MT polymerization?) is required to push the chromosomes beyond the region of mixed MT-polarity, to allow the minus-end forces to take over? Or could CLASP simply provide many “loose”, non-spindle-pole MTs that reduce the opposing plus-end resistance simply by competition? Or is it that MT plus-end directed forces acting between anti-parallel MTs within the bundles eventually shrinks the overlap region to the size depicted in the cartoon? Please provide some suggestion(s) for how this transition might occur*.

We have now added a paragraph to the Discussion to address this issue and propose a number of suggestions.